# A Rescaling-Invariant Lipschitz Bound Based on Path-Metrics for Modern ReLU Network Parameterizations

**Antoine Gonon** [1 2]   **Nicolas Brisebarre** [3]   **Elisa Riccietti** [1]   **Rémi Gribonval** [4]

## Abstract

Robustness with respect to weight perturbations underpins guarantees for generalization, pruning and quantization. Existing guarantees rely on *Lipschitz bounds in parameter space*, cover only plain feed-forward MLPs, and break under the ubiquitous neuron-wise rescaling symmetry of ReLU networks. We prove a new Lipschitz inequality expressed through the $\ell^1$-*path-metric* of the weights. The bound is (i) **rescaling-invariant** by construction and (ii) applies to any ReLU-DAG architecture with any combination of convolutions, skip connections, pooling, and frozen (inference-time) batch-normalization —thus encompassing ResNets, U-Nets, VGG-style CNNs, and more. By respecting the network's natural symmetries, the new bound strictly sharpens prior parameter-space bounds and can be computed in two forward passes. To illustrate its utility, we derive from it a symmetry-aware pruning criterion and show—through a proof-of-concept experiment on a ResNet-18 trained on ImageNet—that its pruning performance matches that of classical magnitude pruning, while becoming totally immune to arbitrary neuron-wise rescalings.

## 1. Introduction

An important challenge about neural networks is to upper bound as tightly as possible the distances between the so-called realizations (*i.e.*, the functions implemented by the considered network) $R_\theta$, $R_{\theta'}$ with parameters $\theta, \theta'$ when evaluated at an input vector $x$, in terms of a (pseudo-)-

distance $d(\theta, \theta')$ and a constant $C_x$:

$$\|R_\theta(x) - R_{\theta'}(x)\|_1 \leqslant C_x d(\theta, \theta'). \qquad (1)$$

This controls the robustness of the function $R_\theta$ with respect to changes in the parameters $\theta$, which can be crucially leveraged to derive generalization bounds (Neyshabur et al., 2018) or theoretical guarantees about pruning or quantization algorithms (Gonon et al., 2023). Yet, to the best of our knowledge, such bounds remain relatively little explored in the literature, and existing ones are expressed with $\ell^p$ metrics on parameters (Gonon et al., 2023; Neyshabur et al., 2018; Berner et al., 2020). For example, such a bound is known (Gonon et al., 2023, Theorem III.1 with $p = \infty$ and $q = 1$) with

$$
\begin{aligned}
d(\theta, \theta') &:= \|\theta - \theta'\|_\infty, \\
C_x &:= (W\|x\|_\infty + 1)WL^2R^{L-1},
\end{aligned}
\qquad (2)
$$

in the case of a layered fully-connected neural network $R_\theta(x) = M_L \operatorname{ReLU}(M_{L-1} \ldots \operatorname{ReLU}(M_1 x))$ with $L$ layers, maximal width $W$, and with weight matrices $M_\ell$ having some operator norm bounded by $R$. Moreover, these known bounds are not satisfying for at least two reasons:

- they are **not invariant under neuron-wise rescalings** of the parameters $\theta$ that leave unchanged its realization $R_\theta$. As we will show, this implies that numerical evaluations of such bounds can be arbitrarily large;

- they **only hold for simple fully-connected models organized in layers**, but not for modern networks that include pooling, skip connections, etc.

To circumvent these issues, we leverage the so-called *path-lifting*, a tool that has recently emerged (Stock & Gribonval, 2023; Bona-Pellissier et al., 2022; Marcotte et al., 2023; Gonon et al., 2024a) in the theoretical analysis of modern neural networks with positively homogeneous activations.

**Main contribution.** We introduce a natural (rescaling-invariant) *metric* based on the *path-lifting*, and shows that it indeed yields a *rescaling-invariant upper bound for the distance of two realizations of a network*. Specifically, denoting $\Phi(\theta)$ the path-lifting (a finite-dimensional vector whose

---

[1]ENS de Lyon, CNRS, Inria, Université Claude Bernard Lyon 1, LIP, UMR 5668, 69342, Lyon cedex 07, France [2]Institute of Mathematics, EPFL, Lausanne, Switzerland [3]CNRS, ENS de Lyon, Université Claude Bernard Lyon 1, LIP, UMR 5668, 69342, Lyon cedex 07, France [4]Inria, CNRS, ENS de Lyon, Université Claude Bernard Lyon 1, LIP, UMR 5668, 69342, Lyon cedex 07, France . Correspondence to: Antoine Gonon .

*Proceedings of the 42$^{nd}$ International Conference on Machine Learning*, Vancouver, Canada. PMLR 267, 2025. Copyright 2025 by the author(s).

Table 1: The path-lifting provides an intermediate space between parameters and function spaces.

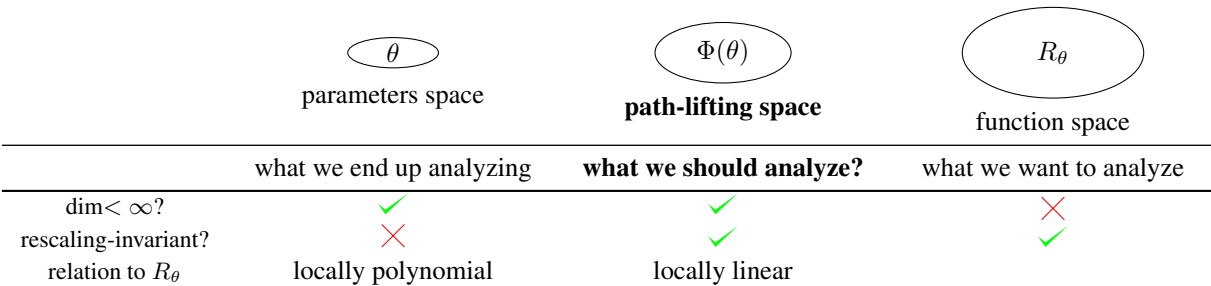

| | what we end up analyzing | what we should analyze? | what we want to analyze |
|---|---|---|---|
| dim$< \infty$? | ✓ | ✓ | ✗ |
| rescaling-invariant? | ✗ | ✓ | ✓ |
| relation to $R_\theta$ | locally polynomial | locally linear | |

definition will be recalled in Section 3) of the network parameters $\theta$, we establish (Theorem 4.1) that for any input $x$, and network parameters $\theta, \theta'$ *with the same entrywise signs*:

$$\|R_\theta(x) - R_{\theta'}(x)\|_1$$
$$\leqslant \max(\|x\|_\infty, 1) \|\Phi(\theta) - \Phi(\theta')\|_1. \quad (3)$$

We call $d(\theta, \theta') := \|\Phi(\theta) - \Phi(\theta')\|_1$ the $\ell^1$-*path-metric*, by analogy with the so-called $\ell^1$-*path-norm* $\|\Phi(\theta)\|_1$, see *e.g.* (Neyshabur et al., 2015; Barron & Klusowski, 2019; Gonon et al., 2024a). Of course, since the $\ell^1$-norm is the largest $\ell^q$-norm ($q \geqslant 1$), this also implies the same inequality for any $\ell^q$-norm on the left-hand side. Besides being intrinsically rescaling-invariant, Inequality (3) holds for the very same general neural network model as in Gonon et al. (2024a) that encompasses pooling, skip connections and so on. This solves the two problems mentioned above and improves on Equation (2). Finally, we show that, under conditions that hold in practical pruning and quantization scenarios, the path-metric is easy to compute in two forward passes, and we provide the corresponding `pytorch` implementation.

Our main theoretical finding, Inequality (3), together with the known properties of $\Phi$ (Gonon et al., 2024a) confirms that the path-lifting $\Phi$ provides an intermediate space between the parameter space and the function space, that shares some advantages of both, see Table 1.

**Plan.** Section 2 places our contribution in context. Section 3 recalls the path-lifting framework of Gonon et al. (2024a) and the notational tools we will use. Section 4 presents our central result—a rescaling-invariant Lipschitz bound expressed through the $\ell^1$-path-metric (Theorem 4.1)—and explains how it sharpens existing bounds and can be computed in two forward passes. Finally, Section 5 shows how the new bound yields a symmetry-aware pruning criterion, shown to match in a proof-of-concept experiment the accuracy of magnitude pruning, while becoming totally immune to neuron-wise rescalings.

## 2. Related Work

Understanding how small weight changes affect a network's output is crucial, e.g., for pruning, quantization, or general-

ization error control. We review these three different use of *parameter–space Lipschitz bounds* in Section 2.1, and then highlight in Section 2.2 how our new, rescaling-invariant bound (Theorem 4.1) interfaces with recent notions of scale-invariant sharpness.

### 2.1. Parameter-Space Lipschitz Bounds in Practice

Parameter–space Lipschitz (or "perturbation" / "sensitivity") bounds already underpin several practical guarantees, but prior results are restricted to plain MLPs and ignore rescaling symmetry.

**(i) Pruning.** Provable pruning schemes quantify how much the output drifts when weights are set to zero. (Liebenwein et al., 2020) and (Baykal et al., 2019) derive such guarantees from layer-wise—but rescaling-*dependent*—Lipschitz constants, and the same mechanism underlies Theorem 5.4 of (Baykal et al., 2022). Our Theorem 4.1 offers an architecture-agnostic, symmetry-aware alternative; Section 5 illustrates this on a ResNet-18.

**(ii) Quantization.** Bounding the error induced by weight rounding likewise depends on how the network reacts to small parameter perturbations. Gonon et al. (2023) provide such bounds for fully-connected nets, while Zhang et al. (2023) and Lybrand & Saab (2021) control the error at the neuron level. Extending those guarantees to CNNs, ResNets or U-Nets requires a global, symmetry-invariant Lipschitz constant—precisely what we provide in Theorem 4.1.

**(iii) Generalization via covering numbers.** Several compression-style analyses (e.g., Arora et al., 2018; Bartlett et al., 2017; Schnoor et al., 2021) follow two steps: (1) a *parameter-space* Lipschitz bound shows that the $\varepsilon$-ball around a weight vector $\theta$ maps into an $\varepsilon'$-ball around its realization $R_\theta$ in function space, yielding an upper bound on that function-space covering number; (2) this covering bound is plugged into Dudley's entropy integral to obtain a Rademacher-complexity (and thus generalization) bound. Because our Lipschitz constant is rescaling-invariant and holds for modern DAG networks, the same pipeline runs without restricting to MLPs and without the looseness intro-

duced when one first factors out rescaling symmetries.

Across pruning, quantization and generalization, two limitations of previous parameter-space bounds—*lack of rescaling-invariance* and *restriction to plain MLPs*—are precisely the issues addressed by Theorem 4.1.

## 2.2. Relation to Scale-Invariant Sharpness

Sharpness metrics (Tsuzuku et al., 2020; Rangamani et al., 2021; Kwon et al., 2021; Wen et al., 2023; Andriushchenko et al., 2023) measure how much the *loss* increases under parameter perturbations, often normalizing those perturbations to remove rescaling dependencies. Our perspective is complementary: we directly bound the *output change* $\|R_\theta(x) - R_{\theta'}(x)\|_1$, independent of any loss or data distribution. As shown in Section 4.4, whenever the loss $\mathcal{L}(\hat{y}, y)$ is Lipschitz in its first argument (e.g., cross-entropy or MSE on a compact domain), Theorem 4.1 yields an immediate upper bound on several scale-invariant sharpness definitions, thus providing a loss-agnostic control over the same perturbation neighborhoods.

## 3. ReLU **DAGs, Invariances, and Path-Lifting**

The neural network model we consider generalizes and unifies several models from the literature, including those from Neyshabur et al. (2015); Kawaguchi et al. (2017); DeVore et al. (2021); Bona-Pellissier et al. (2022); Stock & Gribonval (2023), as detailed in Gonon et al. (2024a, Definition 2.2). This model allows for any Directed Acyclic Graph (DAG) structure that combines standard layers—max-pooling, average-pooling, skip connections, convolution, and (inference-time / frozen form) batch normalization—thereby covering modern networks such as ResNets, VGGs, AlexNet, and many others. The complete formal definition appears in Appendix A.[1]

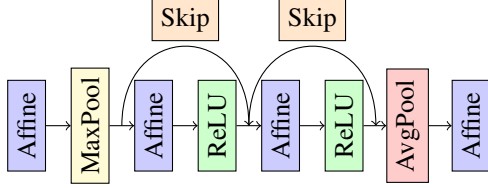

Figure 1: A network with the same ingredients as a ResNet.

---

[1]This DAG-ReLU framework does not cover (i) attention mechanisms and (ii) normalization layers that are not rescaling-invariant (e.g., layer normalization, group normalization). Batch normalization is covered because at inference time its statistics are fixed, so it behaves as an affine layer and remains compatible with the path-lifting framework of (Gonon et al., 2024a).

## 3.1. Rescaling Symmetries.

All network parameters (weights and biases) are gathered in a parameter vector $\theta$, and we denote $R_\theta(x)$ the output of the network when evaluated at input $x$ (the function $x \mapsto R_\theta(x)$ is the so-called *realization* of the network with parameters $\theta$). Due to positive-homogeneity of the ReLU function $t \to \text{ReLU}(t) := \max(0, t)$, in the simple case of a single neuron with no bias we have $R_\theta(x) = v \max(0, \langle u, x \rangle)$ with $\theta = (u, v)$, and for any $\lambda > 0$, the "rescaled" parameter $\tilde{\theta} = (\lambda u, \frac{v}{\lambda})$ implements the same function $R_{\tilde{\theta}} = R_\theta$. A similar rescaling-invariance property holds for the general model of (Stock & Gribonval, 2023; Gonon, 2024) leading to the notion of rescaling-equivalent parameters, denoted $\tilde{\theta} \sim \theta$, which still satisfy $R_{\tilde{\theta}} = R_\theta$.

**Need for rescaling-invariant Lipschitz bounds.** Consider our initial problem of finding a pseudo-metric $d(\theta, \theta')$ and a constant $C_x$ for any input $x$, for which (1) holds. The left hand-side of (1) is invariant under rescaling-symmetries: if $\tilde{\theta} \sim \theta$ then $\|R_{\tilde{\theta}}(x) - R_{\theta'}(x)\|_1 = \|R_\theta(x) - R_{\theta'}(x)\|_1$. However, when $d(\cdot, \cdot)$ is based on a standard $\ell^p$ norm, the right hand-side of (1) is *not* invariant, and in fact $\sup_{\tilde{\theta} \sim \theta} \|\tilde{\theta} - \theta'\|_p = +\infty$, so the bound can in fact be arbitrarily pessimistic:

$$\sup_{\tilde{\theta} \sim \theta} \frac{d(\tilde{\theta}, \theta')}{\|R_{\tilde{\theta}}(x) - R_{\theta'}(x)\|_1} = \infty.$$

Although in general one could *make a bound such as* (1) *invariant* by considering the infimum

$$\inf_{\tilde{\theta} \sim \theta, \tilde{\theta}' \sim \theta'} d(\tilde{\theta}, \tilde{\theta}'),$$

this infimum may be difficult to compute in practice. Therefore, a "good" bound should ideally be both invariant under rescaling symmetries and easy to compute. Invariance to rescaling symmetries is precisely the motivation for the introduction of the path-lifting.

## 3.2. Path-Lifting $\Phi$ and Path-Activation Matrix $A$

**Background.** The *path-lifting* map $\Phi$ and its associated $\ell^1$-*path-norm* were introduced to equip ReLU networks with a coordinate system that is invariant under neuron-wise rescaling. This construction has enabled advances in identifiability (Stock & Gribonval, 2023; Bona-Pellissier et al., 2022), analysis of training dynamics (Marcotte et al., 2023), input-space Lipschitz bounds (Gonon et al., 2024a), and (PAC–Bayes and Rademacher) generalization guarantees (Neyshabur et al., 2015; Gonon et al., 2024a).

This paper does not redefine the path-lifting but *leverages* it to derive, for the first time, a rescaling-invariant parameter-space Lipschitz bound that holds for general DAG-ReLU architectures.

**Definitions (informal).** Given network parameters $\theta$ and an input $x$, we consider two objects from Gonon et al. (2024a, Definition A.1): the path-lifting vector $\Phi(\theta)$ and the path-activation matrix $A(\theta, x)$. Below we give a simplified description sufficient for understanding our main results; full definitions are deferred to Appendix A.

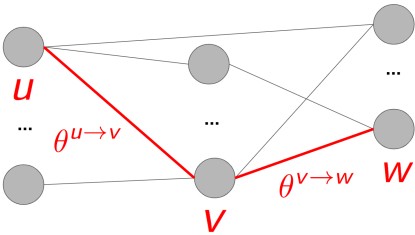

Figure 2: The path-lifting coordinate $\Phi_p(\theta)$ for the path $p = u \to v \to w$ is the product of the weights along that path: $\Phi_p(\theta) = \theta^{u \to v} \theta^{v \to w}$.

The vector $\Phi(\theta) \in \mathbb{R}^{\mathcal{P}}$ is indexed by the set $\mathcal{P}$ of *paths* in the network—i.e., sequences of neurons from an input to an output. For each path, its coordinate in $\Phi(\theta)$ is simply the product of the weights along that path (ignoring nonlinearities). For example, if $p = u \to v \to w$ is a path starting from an input neuron $u$ and ending at an output neuron $w$, and if $\theta_{a \to b}$ denotes the weight on edge $a \to b$, then $\Phi_p(\theta) = \theta^{u \to v} \theta^{v \to w}$, as illustrated in Figure 2.

The path-activation matrix $A(\theta, x) \in \{0, 1\}^{\mathcal{P} \times d_{\text{in}}}$ encodes the information about non-linearities, storing which paths are *active* (i.e., all ReLUs along them are on) for a given input $x$. Entry $A_{p,u}(\theta, x) = 1$ if path $p$ starts at input coordinate $u$ and all neurons along $p$ are activated.

In networks with biases, additional paths starting from hidden neurons are included in $\mathcal{P}$, and $A(\theta, x)$ is extended to $\{0, 1\}^{\mathcal{P} \times (d_{\text{in}}+1)}$ to include bias contributions.

**Key properties of** $(\Phi, A)$**.** These two objects enjoy the following critical features:

- $\Phi(\theta)$ is a vector of monomials in the weights.
- $A(\theta, x)$ is a binary, piecewise-constant matrix of $(\theta, x)$.
- Both $\Phi(\theta)$ and $A(\theta, x)$ are **rescaling-invariant**: if $\tilde{\theta} \sim \theta$ (i.e., $\theta$ and $\tilde{\theta}$ only differ by neuron-wise rescaling, leaving $R_\theta = R_{\tilde{\theta}}$ unchanged), then $\Phi(\tilde{\theta}) = \Phi(\theta)$ and $A(\tilde{\theta}, x) = A(\theta, x)$ for all $x$ (Gonon, 2024, Theorem 2.4.1).
- The network output can be recovered directly from these quantities. For scalar-valued outputs $R_\theta(x)$:

$$R_\theta(x) = \left\langle \Phi(\theta), A(\theta, x) \begin{pmatrix} x \\ 1 \end{pmatrix} \right\rangle, \qquad (4)$$

and a similar form holds for vector-valued networks (Gonon et al., 2024a, Theorem A.1).

**Example (one-hidden-layer network).** Consider a one-hidden-layer ReLU network without bias, with parameters $\theta = (u_1, \ldots, u_k, v_1, \ldots, v_k)$ where $u_i \in \mathbb{R}^{d_{\text{in}}}$, $v_i \in \mathbb{R}^{d_{\text{out}}}$, and realization

$$R_\theta(x) = \sum_{i=1}^{k} \max(0, \langle x, u_i \rangle) v_i \in \mathbb{R}^{d_{\text{out}}}.$$

Then, the path-lifting is

$$\Phi(\theta) = (u_i v_i^\top)_{i \in \{1, \ldots, k\}} \in \mathbb{R}^{k d_{\text{in}} d_{\text{out}}}.$$

The path-activation matrix is

$$A(\theta, x) = \mathbf{I}_{d_{\text{in}}} \otimes (\mathbb{1}_{\langle x, u_i \rangle > 0})_{i=1}^{k} \otimes \mathbf{1}_{d_{\text{out}}},$$

concatenated with a zero column (no biases here). Here, $\mathbf{I}_d$ is the $d \times d$ identity matrix, and $\mathbf{1}_d$ (resp. $\mathbf{0}_d$) is the vector of ones (resp. zeros) of size $d$.

It is straightforward to verify that both $\Phi(\theta)$ and $A(\theta, x)$ remain unchanged under the neuron-wise rescaling $\theta \mapsto \lambda \diamond \theta$, defined by $(v_i, u_i) \mapsto (\frac{1}{\lambda_i} v_i, \lambda_i u_i)$ for any $\lambda \in (\mathbb{R}_{>0})^k$. This transformation leaves the function $R_\theta$ unchanged, i.e., $R_\theta = R_{\lambda \diamond \theta}$ (Gonon et al., 2024a).

## 4. A Rescaling Invariant Lipschitz Bound

Our main result, Theorem 4.1, is a Lipschitz bound with respect to the parameters of the network, as opposed to widespread Lipschitz bounds with respect to the inputs. It precisely proves that (1) holds with a rescaling-invariant pseudo-distance (called the $\ell^1$-path metric) defined via $\Phi$ as $d(\theta, \theta') := \|\Phi(\theta) - \Phi(\theta')\|_1$ and $C_x = \max(\|x\|_\infty, 1)$.

**Theorem 4.1.** *Consider a ReLU DAG neural network, corresponding to an arbitrary DAG network with max-pool etc. as in Section 3, see Figure 1 for an illustration and Definition A.2 in the appendix for a precise definition. Consider parameters vectors $\theta, \theta'$. If for every coordinate $i$, it holds $\theta_i \theta'_i \geqslant 0$, then for every input $x$:*

$$\begin{aligned} \|R_\theta(x) - R_{\theta'}(x)\|_1 \\ \leqslant \max(\|x\|_\infty, 1) \|\Phi(\theta) - \Phi(\theta')\|_1. \quad (5) \end{aligned}$$

*Moreover, for every such neural network architecture, there are non-negative parameters $\theta \neq \theta'$ and a non-negative input $x$ such that Inequality (5) is an equality.*

Since $\|\cdot\|_q \leqslant \|\cdot\|_1$ for any $q \geqslant 1$, Inequality (5) implies the same bound with the $\ell^q$-norm on the left hand-side.

We sketch the proof in Section 4.5. The complete proof is in Appendix B – we actually prove something slightly stronger, but we stick here to Inequality (5) for simplicity.

As discussed in Section 2.1, the parameter-space Lipschitz bound (5), like any such bound, can be incorporated into various pipelines—either to establish theoretical guarantees or to guide practical methods (e.g., algorithms that

minimize these bounds), with applications to pruning, quantization, or generalization. In Section 5, we will focus on pruning. Regarding generalization, let us briefly note that this bound can be used to derive a Rademacher complexity bound for the class of functions $\mathcal{F} := \{R_\theta, \|\Phi(\theta)\|_1 \leqslant r\} = \bigcup_{\text{signs } s}\{R_\theta, \|\Phi(\theta)\|_1 \leqslant r, \text{sgn}(\theta) = s\}$. To bound this complexity, Dudley's integral reduces the task to bounding the covering numbers of each fixed-sign sub-ball $\{R_\theta, \|\Phi(\theta)\|_1 \leqslant r, \text{sgn}(\theta) = s\}$. The inequality (5) enables exactly this, by linking the covering numbers of these function classes to those of the corresponding finite-dimensional sets $\{\Phi(\theta) : \text{sgn}(\theta) = s, \|\Phi(\theta)\|_1 \leqslant r\}$. A full derivation of this approach can be found in Theorem 4.3.1 of (Gonon, 2024). That said, the resulting (Rademacher) generalization bounds are typically looser—by a factor of roughly $\sqrt{\#\text{params}}$—than those of Gonon et al. (2024a), who also leverage the path-norm but through a more refined analysis.

In the rest of this section, we discuss the assumptions of the theorem, the practical computation of the bound and the positioning with respect to previously established Lipschitz bounds.

### 4.1. Why the same–sign assumption is necessary

**A hard impossibility (new contribution).** Let us highlight that the condition $\theta_i \theta_i' \geqslant 0 \; \forall i$ in Theorem 4.1 is *not* a technical convenience. We exhibit in Figure 6 (Appendix B) a minimalistic ReLU network for which *no* finite constant $C_x$ can satisfy (1) once two weights change sign. By prepending and appending arbitrary sub-networks to that minimal counter-example, one gets families where *all but two* edges keep their sign, yet the same divergence occurs. This impossibility shows that *every* rescaling-invariant parameter-space Lipschitz bound based on the path-lifting must, at a minimum, control sign changes. We are not aware of a prior formal statement of this theoretical impossibility.

**Practical relevance.** Many real-world workflows preserve the signs: pruning, uniform quantization, and small SGD steps preserve them; locally, any non-zero $\theta$ admits an $\ell_\infty$ ball where signs are fixed. When occasional flips do occur, Theorem 4.1 remains useful as a local building block: one may use it on each fixed-sign quadrant individually and then *glue* the results established on each quadrant together —exactly the strategy evoked for covering-number generalization proofs (see the discussion after Theorem 4.1).

### 4.2. Approximation and exact computation of $\ell^1$-path-metrics

Since $\Phi(\theta)$ is a vector of combinatorial dimension (it is indexed by paths), it would be intractable to compute the $\ell^1$-path metric $\|\Phi(\theta) - \Phi(\theta')\|_1$ by direct computation of the vector $\Phi(\theta) - \Phi(\theta')$. In this section we investigate efficient *and rescaling-invariant* approximations of the $\ell^1$-

path-metric that turn out to yield exact implementations in cases of practical interest.

A key fact on which the approach is built is that the $\ell^1$-path-norm can be computed in one forward pass (Gonon et al., 2024a). Since, by the lower triangle inequality, we have

$$\left| \|\Phi(\theta)\|_1 - \|\Phi(\theta')\|_1 \right| \leqslant \|\Phi(\theta) - \Phi(\theta')\|_1, \quad (6)$$

the left-hand side of (6) serves as an approximation that can be computed in two forward passes of the network[2].

As we now show, this is an *exact evaluation* of the $\ell^1$-path-metric under practical assumptions, and completed by a rescaling-invariant upper bound (cf. Inequality (8) below).

**Lemma 4.2.** *Inequality* (6) *is an equality as soon as* $|\Phi(\theta)| \geqslant |\Phi(\theta')|$ coordinatewise: *in this case we have*

$$\|\Phi(\theta) - \Phi(\theta')\|_1 = \|\Phi(\theta)\|_1 - \|\Phi(\theta')\|_1. \quad (7)$$

*Proof.* For vectors $a, b$ with $|a_i| \geqslant |b_i|$ for every $i$, we have

$$\|a\|_1 - \|b\|_1 = \sum_i |a_i| - |b_i| = \sum_i |a_i - b_i| = \|a - b\|_1.$$

$\square$

An important scenario where $|\Phi(\theta)| \geqslant |\Phi(\theta')|$ indeed holds is when $|\theta| \geqslant |\theta'|$ coordinatewise. **The latter is true in at least two significant situations**: when $\theta'$ is obtained from $\theta$ by **pruning**, or through **quantization** provided that rounding is done either systematically towards zero or systematically away from zero.

Note that $|\theta| \geqslant |\theta'|$ is not the only situation where $|\Phi(\theta)| \geqslant |\Phi(\theta')|$. For instance, due to the rescaling-invariance of $\Phi(\cdot)$, if $\tilde{\theta}$ is rescaling-equivalent to $\theta$ the coordinatewise inequality $|\Phi(\tilde{\theta})| \geqslant |\Phi(\theta')|$ remains valid, even though in general such a $\tilde{\theta}$ non longer satisfies $|\tilde{\theta}| \geqslant |\theta'|$ coordinatewise.

Even out of such practical scenarios, the $\ell^1$-path-metric also satisfies an *invariant* upper bound.

**Lemma 4.3** (Informal version of Lemma F.3)**.** *Consider a DAG ReLU network with $L$ layers and width $W$. For any parameter $\theta$, denote by $\text{N}(\theta)$ its normalized version, deduced from $\theta$ by applying rescaling-symmetries such that each neuron has its vector of incoming weights equal to $1$, except for output neurons. It holds for all parameters $\theta, \theta'$:*

$$\|\Phi(\theta) - \Phi(\theta')\|_1$$
$$\leqslant (W^2 + \min(\|\Phi(\theta)\|_1, \|\Phi(\theta')\|_1) \cdot LW) \|\text{N}(\theta) - \text{N}(\theta')\|_\infty. \quad (8)$$

---

[2]For a ResNet18, we timed it to 15ms. Specifically, we timed the function `get_path_norm` available at `github.com/agonon/pathnorm_toolkit` using `pytorch.utils.benchmark`. Experiments were made on an NVIDIA GPU A100-40GB, with processor AMD EPYC 7742 64-Core.

The proof is in Appendix F. In all the cases of interest we consider, the lower bound (6) is exact as a consequence of Lemma 4.2. We leave it to future work to compare the lower bound with the upper bound of Lemma 4.3 in specific cases where the lower bound is inexact.

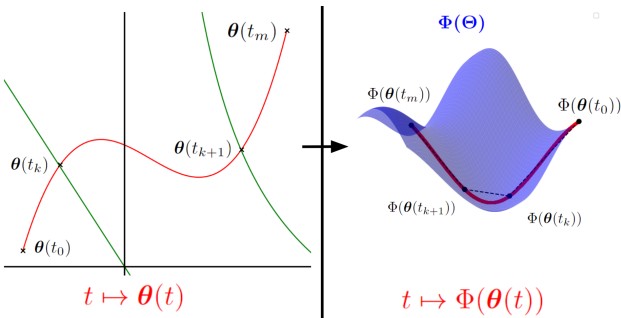

Figure 3: Illustration of the proof of Theorem 4.1, see Section 4.5 for an explanation.

### 4.3. Improvement over previous Lipschitz bounds

Inequality (5) improves on the Lipschitz bound (1) specified with Equation (2), as the next result shows.

**Lemma 4.4.** *Consider a simple layered fully-connected neural network architecture with $L \geqslant 1$ layers, corresponding to functions $R_\theta(x) = M_L \operatorname{ReLU}(M_{L-1} \ldots \operatorname{ReLU}(M_1 x))$ with each $M_\ell$ denoting a matrix, and parameters $\theta = (M_1, \ldots, M_L)$. For a matrix $M$, denote by $\|M\|_{1,\infty}$ the maximum $\ell^1$-norm of a row of $M$. Consider $R \geqslant 1$ and define the set $\Theta$ of parameters $\theta = (M_1, \ldots, M_L)$ such that $\|M_\ell\|_{1,\infty} \leqslant R$ for every $\ell \in [\![1, L]\!]$. Then, for every parameters $\theta, \theta' \in \Theta$*

$$\|\Phi(\theta) - \Phi(\theta')\|_1 \leqslant L W^2 R^{L-1} \|\theta - \theta'\|_\infty. \quad (9)$$

*Moreover the right hand-side can be arbitrarily worse that the $\ell^1$-pseudo-metric in the left hand side: over all rescaling-equivalent parameters $\tilde{\theta} \sim \theta$, it holds*

$$\sup_{\tilde{\theta} \sim \theta} \frac{\|\tilde{\theta} - \theta'\|_\infty}{\|\Phi(\tilde{\theta}) - \Phi(\theta')\|_1} = \infty.$$

The proof of Lemma 4.4 is in Inequality (23) in Appendix G.

The *invariant* Lipschitz bound (5) combined with (9) yields a (non-invariant) bound on $\|R_\theta(x) - R_{\theta'}(x)\|_1$:

$$\max(\|x\|_\infty, 1) L W^2 R^{L-1} \|\theta - \theta'\|_\infty.$$

In comparison the generic bound (1) specified with (2) reads

$$(W\|x\|_\infty + 1) W L^2 R^{L-1} \|\theta - \theta'\|_\infty.$$

As soon as $\|x\|_\infty \geqslant 1$ the latter is a looser bound than the former.

### 4.4. Implication for scale-invariant sharpness

Let $\ell : \mathbb{R}^{d_{\text{out}}} \times \mathbb{R}^{d_{\text{out}}} \to \mathbb{R}_+$ be $\kappa$-Lipschitz in its first argument with respect to $\ell^1$-norm, and assume a data distribution $\mathcal{D}$ over $(x, y)$. For any parameter $\theta$ and perturbation radius $\rho > 0$, consider the *scale-adaptive worst-case sharpness* (see Definition 2 in Kwon et al. (2021), or Equation 1 in Andriushchenko et al. (2023)):

$$\operatorname{Sharp}_\rho(\theta) :=$$
$$\sup_{\|\delta \odot |\theta|^{-1}\|_p \leqslant \rho} \mathbb{E}_{(x,y)\sim\mathcal{D}}\Big(\ell(R_{\theta+\delta}(x), y) - \ell(R_\theta(x), y)\Big)$$

**Lemma 4.5.** *For every $\rho \in (0, 1)$ and every $\theta$,*

$$\operatorname{Sharp}_\rho(\theta) \leqslant \kappa \mathbb{E}_{x\sim\mathcal{D}_x}\big[\max(\|x\|_\infty, 1)\big]$$
$$\sup_{\|\delta \odot |\theta|^{-1}\|_p \leqslant \rho} \|\Phi(\theta + \delta) - \Phi(\theta)\|_1$$

*Proof.* Lipschitzness of the loss yields $\ell(R_{\theta+\delta}(x), y) - \ell(R_\theta(x), y) \leqslant \kappa\|R_{\theta+\delta}(x) - R_\theta(x)\|_1$. The condition $\|\delta \odot |\theta|^{-1}\|_p \leqslant \rho$ implies $\|\delta \odot |\theta|^{-1}\|_\infty \leqslant \rho$. Thus $\delta_i \leqslant |\theta_i|\rho < |\theta_i|$ for every coordinate $i$ and we get $\operatorname{sgn}(\theta_i + \delta_i) = \operatorname{sgn}(\theta_i)$. Therefore Theorem 4.1 applies and gives $\|R_{\theta+\delta}(x) - R_\theta(x)\|_1 \leqslant \max(\|x\|_\infty, 1)\|\Phi(\theta + \delta) - \Phi(\theta)\|_1$. $\square$

Lemma 4.5 shows that our path-metric controls the scale-adaptive sharpness notions used, e.g., in (Kwon et al., 2021) and (Andriushchenko et al., 2023).

### 4.5. Proof sketch of Theorem 4.1 (full proof in Appendix B)

Given an input $x$, the proof of Theorem 4.1 consists in defining a trajectory $t \in [0, 1] \to \theta(t) \in \Theta$ (**red** curve in Figure 3) that starts at $\theta$, ends at $\theta'$, and with finitely many breakpoints $0 = t_0 < t_1 < \cdots < t_m = 1$ such that the path-activations $A(\theta(t), x)$ are constant on the open intervals $t \in (t_k, t_{k+1})$. Each breakpoint corresponds to a value where the activation of at least one path (hence at least one neuron) changes in the neighborhood of $\theta(t)$. For instance, in the left part of Figure 3, the straight **green** line (resp. quadratic **green** curve) corresponds to a change of activation of a ReLU neuron (for a given input $x$ to the network) in the first (resp. second) layer.

With such a trajectory, given the key property (4), each quantity $|R_{\theta(t_k)}(x) - R_{\theta(t_{k+1})}(x)|$ can be controlled in terms of $\|\Phi(\theta(t_k)) - \Phi(\theta(t_{k+1}))\|_1$, and if the path is "nice enough", then this control can be extended globally from $t_0$ to $t_m$.

There are two obstacles: 1) proving that there are finitely many breakpoints $t_k$ as above (think of $t \mapsto t^{n+2} \sin(1/t)$ that is $n$-times continuously differentiable but still crosses $t = 0$ an infinite number of times around zero), and 2)

proving that the length $\sum_{k=1}^{m} \|\Phi(\theta(t_k)) - \Phi(\theta(t_{k+1}))\|_1$ of the broken line with vertices $\Phi(\theta(t_k))$ (dashed line on the right part of Figure 3) is bounded from above by $\|\Phi(\theta) - \Phi(\theta')\|_1$ times a reasonable factor. Trajectories satisfying these two properties are called "admissible" trajectories.

The first property is true as soon as the trajectory $t \mapsto \theta(t)$ is smooth enough (analytic, say). For this, we will notably exploit that the output of a ReLU neuron in the $d$-th layer of a layered fully-connected network is a piecewise polynomial function of the parameters $\theta$ of degree at most $d$ (Gonon et al., 2024a, consequence of Lemma A.1), (Bona-Pellissier et al., 2022, consequence of Propositions 1 and 2). The property second is true *with factor one* thanks to a monotonicity property of the chosen trajectory.

The core of the proof consists in exhibiting a trajectory with these two properties. To the best of our knowledge, the proof of Inequality (3) is the first to *practically leverage the idea of "adequately navigating" through the different regions in $\theta$ where the network is polynomial*[3] by respecting the geometry induced by $\Phi$, see Figure 3 for an illustration.

# 5. Rescaling-Invariant Pruning

We exploit Inequality (3) to design a pruning rule that is *both* effective and invariant to neuron-wise rescaling. Instead of ranking weights by their magnitude, we rank them by their $\ell^1$-*path-metric* contribution. We show in a proof-of-concept on a ResNet-18 trained on ImageNet-1k under the lottery-ticket "rewind-and-fine-tune" schedule (Frankle et al., 2020) that this *path-magnitude* rule achieves the same accuracy as classical magnitude pruning while becoming totally immune to arbitrary rescalings.

## 5.1. Pruning: a quick overview

Pruning typically involves ranking weights by a chosen criterion and removing (setting to zero) those deemed less important (Han et al., 2016). Early criteria considered either weight magnitudes (Hanson & Pratt, 1988; Han et al., 2016) or the loss's sensitivity to each weight (LeCun et al., 1989; Hassibi & Stork, 1992). Building on these foundations, more sophisticated pruning methods have emerged, often formulated as complex optimization problems solved via advanced algorithms. For example, consider the *entrywise* loss's sensitivity criterion of (LeCun et al., 1989). In principle, all the costs should be recomputed after each pruning decision, since removing one weight affects the costs of the others. A whole literature focuses on turning the cost of (LeCun et al., 1989) into an algorithm that would take

---

[3]The mapping $(\theta, x) \mapsto R_\theta(x)$ is indeed known to be piecewise polynomial in the coordinates of $\theta$ (Gonon et al., 2024a, consequence of Lemma A.1)(Bona-Pellissier et al., 2022, consequence of Propositions 1 and 2).

into account these global dependencies (Singh & Alistarh, 2020; Yu et al., 2022; Benbaki et al., 2023). This line of work recently culminated in CHITA (Benbaki et al., 2023), a pruning approach that scales up to millions of parameters through substantial engineering effort.

Here, we introduce a *path-magnitude* cost defined for each *individual weight* but that depends on the *global* configuration of the weights. Just as sensitivity-based costs (LeCun et al., 1989), these costs should in principle be re-computed after each pruning decision. While taking these global dependencies into account is expected to provide better performance, this is also expected to require a huge engineering effort, similar to what has been done in (Singh & Alistarh, 2020; Yu et al., 2022; Benbaki et al., 2023), which is beyond the scope of this paper. Our goal here is more modest: we aim at providing a simple proof-of-concept to show the promises of the path-lifting for *rescaling-invariant* pruning.

**Notion of pruned parameter**. Considering a neural network architecture given by a graph $G$, we use the shorthand $\mathbb{R}^G$ to denote the corresponding set of parameters (see Definition A.2 for a precise definition). By definition, a pruned version $\theta'$ of $\theta \in \mathbb{R}^G$ is a "Hadamard" product $\theta' = s \odot \theta$, where $s \in \{0,1\}^G$ and $\|s\|_0$ is "small". We denote $\mathbf{1}_G \in \mathbb{R}^G$ the vector filled with ones, $e_i \in \mathbb{R}^G$ the $i$-th canonical vector, $s_i := \mathbf{1}_G - e_i$, and introduce the specialized notation $\theta_{-i} := s_i \odot \theta$ for the vector where a single entry (the weight of an edge or the bias of a hidden or output neuron) of $\theta$, indexed by $i$, is set to zero.

## 5.2. Proposed rescaling-invariant pruning criterion

The starting point of the proposed pruning criterion is that, given any $\theta$, the pair $\theta, \theta'$ with $\theta' := s \odot \theta$ satisfies the assumptions of Theorem 4.1, hence for all input $x$ we have $|R_\theta(x) - R_{\theta'}(x)| \leqslant \|\Phi(\theta) - \Phi(\theta')\|_1 \max(1, \|x\|_\infty)$. Specializing this observation to the case where a single entry (the weight of an edge, or the bias of hidden or output neuron indexed by $i$) of $\theta$ is pruned (*i.e.*, $\theta' = \theta_{-i}$) suggests the following definition, which will serve as a *pruning criterion*:

**Definition 5.1.** We denote

$$\texttt{Path-Mag}(\theta, i) := \|\Phi(\theta) - \Phi(\theta_{-i})\|_1. \quad (10)$$

This measures the contribution to the path-norm of all paths $p$ containing entry $i$: when $i \notin p$ we have $\Phi_p(\theta_{-i}) = \Phi_p(\theta)$, while otherwise $\Phi_p(\theta_{-i}) = 0$. Since $\theta$ and $\theta_{-i}$ satisfy the assumptions of Lemma 4.2 we have

$$\texttt{Path-Mag}(\theta, i) \underset{(7)}{=} \|\Phi(\theta)\|_1 - \|\Phi(\theta_{-i})\|_1 \quad (11)$$

$$= \sum_{p \in \mathcal{P}} |\Phi_p(\theta)| - \sum_{p \in \mathcal{P}: i \notin p} |\Phi_p(\theta)|$$

$$= \sum_{p \in \mathcal{P}: i \in p} |\Phi_p(\theta)| \quad (12)$$

Table 2: Comparison of pruning criteria across key properties. Being *data-specific* or *loss-specific* can be both a strength (leveraging the training loss and data for more accurate pruning) and a limitation (requiring access to additional information). Being *rescaling-invariant* ensures the pruning mask is unaffected by neuron-wise weight rescaling.

| Criterion | Rescaling-Invariant | Error bound | Data-Specific | Loss-Specific | Efficient to Compute | Versatile[a] |
|---|---|---|---|---|---|---|
| Magnitude | No | Yes – (1)-(2) | No | No | Yes | Yes |
| Loss-Sensitivity (Taylor Expansion) | Yes in theory Not in practice[c] | No | Yes | Yes | Depends[b] | Yes |
| **Path-Magnitude** | Yes | Yes – (13) | No | No | Yes | Yes |

[a] Can be used to design greedy approaches (including $\ell^0$-based methods) and supports both structured and unstructured pruning.
[b] Depends on how higher-order derivatives of the loss are taken into account. E.g., using only the diagonal of the Hessian can be relatively quick, but computing the full Hessian is infeasible for large networks. See Table 3 for experiments.
[c] See Equation (16) in Appendix E for invariance in theory, and end of Appendix E for non-invariance in practice.

In light of (5), to limit the impact of pruning on the perturbation of the initial function $R_\theta$, it is natural to choose a coordinate $i$ of $\theta$ leading to a small value of this criterion.

**Lemma 5.2.** `Path-Mag` *enjoys the following properties:*

- **rescaling-invariance**: *for each $\theta \in \mathbb{R}^G$ and index $i$,* `Path-Mag`$(\theta, i) = $ `Path-Mag`$(\tilde{\theta}, i)$ *for every rescaling-equivalent parameters $\tilde{\theta} \sim \theta$;*

- **error bound:** *denote $s := \mathbf{1}_G - \sum_{i \in I} e_i$ where $I$ indexes entries of $\theta \in \mathbb{R}^G$ to be pruned. We have*

$$|R_\theta(x) - R_{s \odot \theta}(x)| \\ \leqslant \Big( \sum_{i \in I} \texttt{Path-Mag}(\theta, i) \Big) \max(1, \|x\|_\infty). \quad (13)$$

- **computation with only two forward passes**: *using Equation (11) and the fact that $\|\Phi(\cdot)\|_1$ is computable in one forward pass (Gonon et al., 2024a).*

- **efficient joint computation** *for all entries***:** *we have*

$$(\texttt{Path-Mag}(\theta, i))_i = \theta \odot \nabla_\theta \|\Phi(\theta)\|_1 \quad (14)$$

*that enables computation via auto-differentiation.*

The proof is given in Appendix C. We summarize these properties in Table 2.

### 5.3. Considered (basic) path-magnitude pruning method

Equipped with `Path-Mag`, a basic rescaling-invariant pruning approach is to minimize the upper-bound (13). This is achieved via simple *reverse* hard thresholding:

1. **Score all weights.** The entire vector $(\texttt{Path-Mag}(\theta, i))_i$ can be produced in *one* reverse-mode autograd pass via Eq. (14).

2. **Prune.** Zero-out the weights with the smallest scores.

To the best of our knowledge, this is the first practical network pruning method that is both *invariant* under rescaling symmetries and *endowed with guarantees* such as (11) on modern networks.

Table 3: Run-time (in milliseconds) to score *all* weights. Time of a forward pass included for reference. Entries in "OBD" and "Forward" columns show times for batch-sizes 1 and 128 (e.g., "13–60" means 13 ms at batch size 1 vs. 60 ms at batch size 128). See Appendix E for details.

| Network | Forward | Mag | OBD | **Path-Mag** |
|---|---|---|---|---|
| AlexNet | 1.7–133 | 0.5 | 13–60 | 14 |
| VGG16 | 2.3–198 | 1.4 | 31–675 | 61 |
| ResNet18 | 3.6–142 | 3.2 | 51–155 | 32 |

While Table 3 shows that path-magnitude pruning is *computationally feasible*, we must also verify that when injected in usual pruning pipelines, it yields *acceptable accuracies*.

### 5.4. Proof-of-concept study

As a simple proof-of-concept, we prune a dense ResNet-18 trained on ImageNet-1k.

**Setup.** Dense ResNet-18 on ImageNet-1k, standard training hyper-parameters, lottery-ticket "rewind-and-fine-tune" schedule (Frankle et al., 2021). We benchmark three pruning criteria: (i) magnitude, (ii) magnitude after a *random* neuron-wise rescaling, (iii) our path-magnitude. See Appendix D for details.

**Results.** Table 4 reports top-1 accuracy after fine-tuning when pruning either 40, 60 or 80% of the weights. Path-magnitude matches[4] magnitude pruning on the un-rescaled network and *completely eliminates* the 5–50% accuracy drop

---

[4]We performed *no* extra hyper-parameter tuning for path-magnitude; we reused the lottery-ticket settings published for magnitude pruning in Frankle et al. (2021).

Table 4: Top-1 ImageNet accuracy (%) on ResNet-18 after one-shot pruning, rewind, and 85-epoch fine-tune. Original accuracy: 67.7%. Three pruning levels shown; more in Appendix D.

| Pruning level | 40% | 60% | 80% |
|---|---|---|---|
| **Path-magnitude** | 68.6 | 67.9 | 66.0 |
| Magnitude | 68.8 | 68.2 | 66.5 |
| Magnitude (rescaled) | 63.1 | 57.5 | 15.8 |

incurred when magnitude is applied after rescaling. Figure 4 shows the full training trajectory at 40 % sparsity.

**Runtime.** Path-magnitude scores for all weights are computed in 32 ms (Table 3), comparable to a single forward pass (see Appendix E for details).

These results confirm that rescaling invariance is not just cosmetic: it prevents large accuracy losses under benign weight re-scalings while keeping the computational cost low. A broader comparison with structured and iterative methods such as CHITA is left for future work.

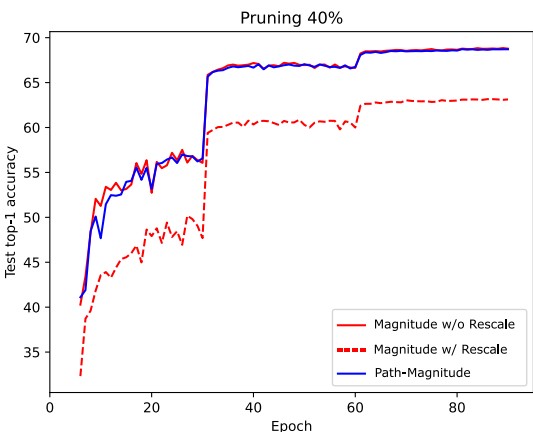

Figure 4: Top-1 accuracy during fine-tuning at 40 % sparsity. Path-magnitude overlaps exactly with itself after random neuron-wise rescaling, while magnitude pruning degrades.

### 5.5. Discussion and possible future extension

The cost `Path-Mag`$(\theta, i)$ is defined per weight, but its value for a given weight indexed by $i$ also depends on the other weights. Therefore, one could hope to achieve better pruning properties if, once a weight is pruned, the path-magnitude costs of the remaining weights were updated. This is reminiscent of the loss-sensitivity cost (LeCun et al., 1989) that associates to each weight $i$ (a surrogate of) the difference $\ell(\theta_{-i}) - \ell(\theta)$, where $\ell$ is a given loss function. The challenge is similar in both cases: how to account for *global* dependencies between the pruning costs associated to each *individual* weight? In this direction, a whole literature

has developed techniques attempting to *globally* minimize (a surrogate of) $\ell(s \odot \theta) - \ell(\theta)$ over the (combinatorial) choice of a support $s$ satisfying an $\ell^0$-constraint. Such approaches have been scaled up to million of parameters in (Benbaki et al., 2023) by combining a handful of clever algorithmic designs. Similar iterative or greedy strategies could be explored to aim at solving the (seemingly) combinatorial $\ell^0$-optimization problem $\|\Phi(s \odot \theta) - \Phi(\theta)\|_1$.

## 6. Conclusion

We introduced a new Lipschitz bound on the distance between two neural network realizations, leveraging the path-lifting framework of Gonon et al. (2024a). By formulating this distance in terms of the $\ell^1$-path-metric, our result applies to a broad class of modern ReLU networks—including ones like ResNets or AlphaGo—and crucially overcomes the arbitrary pessimism arising in non-invariant parameter-based bounds. Beyond providing a theoretical guarantee, we also argued that this metric can be computed efficiently in practical scenarios such as pruning and quantization.

We then demonstrated how to apply path-lifting to pruning: the *path-magnitude* criterion defines a rescaling-invariant measure of the overall contribution of a weight. In a proof-of-concept on a ResNet-18 trained on ImageNet, *path-magnitude* pruning yields an accuracy on par with standard magnitude pruning. This connects the theoretical notion of path-lifting to a practical goal: making pruning decisions that cannot be undermined by mere neuron-wise rescaling.

This work raises several directions for future research. First, a natural challenge is to establish sharper versions of our core result (Theorem 4.1), typically with metrics still based on the path-lifting but using $\ell^p$-norms with $p > 1$, or by deriving functional bounds in expectation (over a given probability distribution of inputs).

Second, more advanced iterative algorithms, akin to second-order pruning techniques, might benefit from path-lifting as a fundamental building block, improving upon the simple one-pass approach used in our proof-of-concept while retaining invariance properties (see Section 5.5).

Finally, although our main theorem improves existing Lipschitz bounds and extends them to a wide range of network architectures, the potential applications of the path-lifting perspective–and its invariance under rescaling–are far from exhausted. Quantization and generalization, in particular, are two important areas where the present findings might stimulate further developments on metrics that offer both theoretical grounding and compelling practical properties.

## Acknowledgements

This work was supported in part by the AllegroAssai ANR-19-CHIA-0009, by the NuSCAP ANR-20-CE48-0014 projects of the French Agence Nationale de la Recherche and by the SHARP ANR project ANR-23-PEIA-0008 in the context of the France 2030 program.

The authors thank the Blaise Pascal Center for the computational means. It uses the SIDUS solution (Quemener & Corvellec, 2013) developed by Emmanuel Quemener.

## Impact Statement

This paper presents work whose goal is to advance the field of Machine Learning. There are many potential societal consequences of our work, none which we feel must be specifically highlighted here.

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

# Appendices

# A. Path-lifting, activations, and a fixed incidence matrix

We recall the construction of Gonon et al. (2024a), but instead of considering the path-activation matrix $A(\theta, x)$ as in Gonon et al. (2024a), we introduce two new objects $A$ and $a(\theta, x)$ that lead to mathematically equivalent formulas but to a lighter proof of Theorem 4.1:

- the *path-activation vector* $a(\theta, x)$, and

- a *fixed* incidence matrix $A$ that depends only on the DAG architecture, never on $\theta$ or $x$.

## A.1. Network architecture

**Definition A.1** (ReLU and $k$-max-pooling activation functions)**.** The ReLU function is defined as $\mathrm{ReLU}(x) := x\mathbb{1}_{x \geqslant 0}$ for $x \in \mathbb{R}$. The $k$-max-pooling function $k\text{-}\mathtt{pool}(x) := x_{(k)}$ returns the $k$-th largest coordinate of $x \in \mathbb{R}^d$.

**Definition A.2** (DAG-ReLU neural network (Gonon et al., 2024a))**.** Consider a Directed Acyclic Graph (DAG) $G = (N, E)$ with edges $E$, and vertices $N$ called neurons. For a neuron $v$, the sets $\mathrm{ant}(v), \mathrm{suc}(v)$ of antecedents and successors of $v$ are $\mathrm{ant}(v) := \{u \in N, u \to v \in E\}, \mathrm{suc}(v) := \{u \in N, v \to u \in E\}$. Neurons with no antecedents (resp. no successors) are called input (resp. output) neurons, and their set is denoted $N_{\mathrm{in}}$ (resp. $N_{\mathrm{out}}$). Neurons in $N \setminus (N_{\mathrm{in}} \cup N_{\mathrm{out}})$ are called hidden neurons. Input and output dimensions are respectively $d_{\mathrm{in}} := |N_{\mathrm{in}}|$ and $d_{\mathrm{out}} := |N_{\mathrm{out}}|$.

• A **ReLU neural network architecture** is a tuple $(G, (\rho_v)_{v \in N \setminus N_{\mathrm{in}}})$ composed of a DAG $G = (N, E)$ with attributes $\rho_v \in \{\mathrm{id}, \mathrm{ReLU}\} \cup \{k\text{-}\mathtt{pool}, k \in \mathbb{N}_{>0}\}$ for $v \in N \setminus (N_{\mathrm{out}} \cup N_{\mathrm{in}})$ and $\rho_v = \mathrm{id}$ for $v \in N_{\mathrm{out}}$. We will again denote the tuple $(G, (\rho_v)_{v \in N \setminus N_{\mathrm{in}}})$ by $G$, and it will be clear from context whether the results depend only on $G = (N, E)$ or also on its attributes. Define $N_\rho := \{v \in N, \rho_v = \rho\}$ for an activation $\rho$, and $N_{*\text{-}\mathtt{pool}} := \cup_{k \in \mathbb{N}_{>0}} N_{k\text{-}\mathtt{pool}}$. A neuron in $N_{*\text{-}\mathtt{pool}}$ is called a $*$-max-pooling neuron. For $v \in N_{*\text{-}\mathtt{pool}}$, its kernel size is defined as being $|\mathrm{ant}(v)|$.

• **Parameters** associated with this architecture are vectors[5] $\theta \in \mathbb{R}^G := \mathbb{R}^{E \cup N \setminus N_{\mathrm{in}}}$. We call bias $b_v := \theta_v$ the coordinate associated with a neuron $v$ (input neurons have no bias), and denote $\theta^{u \to v}$ the weight associated with an edge $u \to v \in E$. We will often denote $\theta^{\to v} := (\theta^{u \to v})_{u \in \mathrm{ant}(v)}$ and $\theta^{v \to} := (\theta^{u \to v})_{u \in \mathrm{suc}(v)}$.

• The **realization** of a neural network with parameters $\theta \in \mathbb{R}^G$ is the function $R_\theta^G : \mathbb{R}^{N_{\mathrm{in}}} \to \mathbb{R}^{N_{\mathrm{out}}}$ (simply denoted $R_\theta$ when $G$ is clear from the context) defined for every input $x \in \mathbb{R}^{N_{\mathrm{in}}}$ as

$$R_\theta(x) := (v(\theta, x))_{v \in N_{\mathrm{out}}},$$

where we use the same symbol $v$ to denote a neuron $v \in N$ and the associated function $v(\theta, x)$, defined as $v(\theta, x) := x_v$ for an input neuron $v$, and defined by induction otherwise

$$v(\theta, x) := \begin{cases} \rho_v(b_v + \sum_{u \in \mathrm{ant}(v)} u(\theta, x)\theta^{u \to v}) & \text{if } \rho_v = \mathrm{ReLU} \text{ or } \rho_v = \mathrm{id}, \\ k\text{-}\mathtt{pool}\left((b_v + u(\theta, x)\theta^{u \to v})_{u \in \mathrm{ant}(v)}\right) & \text{if } \rho_v = k\text{-}\mathtt{pool}. \end{cases}$$

## A.2. Paths and the path-lifting

**Definition A.3** (Paths and depth in a DAG (Gonon et al., 2024a))**.** Consider a DAG $G = (N, E)$ as in Definition A.2. A path of $G$ is any sequence of neurons $v_0, \ldots, v_d$ such that each $v_i \to v_{i+1}$ is an edge in $G$. Such a path is denoted $p = v_0 \to \ldots \to v_d$. This includes paths reduced to a single $v \in N$, denoted $p = v$. The *length* of a path is $\mathtt{length}(p) = d$ (the number of edges). We will denote $p_\ell := v_\ell$ the $\ell$-th neuron for a general $\ell \in \{0, \ldots, \mathtt{length}(p)\}$ and use the shorthand $p_{\mathrm{end}} = v_{\mathtt{length}(p)}$ for the last neuron. The *depth of the graph* $G$ is the maximum length over all of its paths. If $v_{d+1} \in \mathrm{suc}(p_{\mathrm{end}})$ then $p \to v_{d+1}$ denotes the path $v_0 \to \ldots \to v_d \to v_{d+1}$. We denote by $\mathcal{P}^G$ (or simply $\mathcal{P}$) the set of paths ending at an output neuron of $G$.

**Definition A.4** (Sub-graph ending at a given neuron)**.** Given a neuron $v$ of a DAG $G$, we denote $G^{\to v}$ the graph deduced from $G$ by keeping only the largest subgraph with the same inputs as $G$ and with $v$ as a single output: every neuron $u$ with no path to reach $v$ through the edges of $G$ is removed, as well as all its incoming and outcoming edges. We will use the shorthand $\mathcal{P}^{\to v} := \mathcal{P}^{G^{\to v}}$ to denote the set of paths in $G$ ending at $v$.

---

[5]For an index set $I$, denote $\mathbb{R}^I = \{(\theta_i)_{i \in I}, \theta_i \in \mathbb{R}\}$.

**Definition A.5** (Path-lifting $\Phi(\theta)$)**.** Consider a DAG-ReLU neural network $G$ as in Definition A.2 and parameters $\theta \in \mathbb{R}^G$ associated with $G$. For $p \in \mathcal{P}$, define

$$
\Phi_p(\theta) := \begin{cases} \prod_{\ell=1}^{\texttt{length}(p)} \theta^{v_{\ell-1} \to v_\ell} & \text{if } p_0 \in N_{\text{in}}, \\ b_{p_0} \prod_{\ell=1}^{\texttt{length}(p)} \theta^{v_{\ell-1} \to v_\ell} & \text{otherwise,} \end{cases}
$$

where an empty product is equal to 1 by convention. The path-lifting $\Phi^G(\theta)$ of $\theta$ is

$$
\Phi^G(\theta) := (\Phi_p(\theta))_{p \in \mathcal{P}^G}.
$$

This is often denoted $\Phi$ when the graph $G$ is clear from the context. We will use the shorthand $\Phi^{\to v} := \Phi^{G^{\to v}}$ to denote the path-lifting associated with $G^{\to v}$ (Definition A.4).

### A.3. Path-activation vector and fixed incidence matrix

**Definition A.6** (Activation of edges, neurons, and paths)**.** Given $\theta, x$, the activation of an edge $u \to v$ is $a_{u \to v}(\theta, x) := 1$ if $v$ is identity, $\mathbb{1}_{v(\theta,x)>0}$ if $v$ is ReLU, and for $k$-max-pool it is 1 only for the (lexicographically) first antecedent achieving the $k$-th maximum. For a neuron $v$ set $a_v(\theta, x) := 1$ if $v$ is input, identity, or max-pool, and $\mathbb{1}_{v(\theta,x)>0}$ if $v$ is ReLU. For a path $p = v_0 \to \cdots \to v_d$ define

$$
a_p(\theta, x) := a_{v_0}(\theta, x) \prod_{\ell=1}^{d} a_{v_{\ell-1} \to v_\ell}(\theta, x) \ \in \{0, 1\}.
$$

The *path-activation vector* is $a(\theta, x) := (a_p(\theta, x))_{p \in \mathcal{P}} \in \{0, 1\}^{\mathcal{P}}$.

**Definition A.7** (Fixed incidence matrix $A$)**.** Consider a new symbol $v_{\texttt{bias}}$ that is not used to denote neurons. Instead of considering as in (Gonon et al., 2024a) the path-activations matrix $\boldsymbol{A}(\theta, x) \in \mathbb{R}^{\mathcal{P} \times (N_{\text{in}} \cup \{v_{\texttt{bias}}\})}$ whose coordinates are indexed by paths $p \in \mathcal{P}$ and neurons $u \in N_{\text{in}} \cup \{v_{\texttt{bias}}\}$ and are given by

$$
(\boldsymbol{A}(\theta, x))_{p,u} := \begin{cases} a_p(\theta, x) \mathbb{1}_{p_0 = u} & \text{if } u \in N_{\text{in}}, \\ a_p(\theta, x) \mathbb{1}_{p_0 \notin N_{\text{in}}} & \text{otherwise when } u = v_{\texttt{bias}}. \end{cases}
$$

we define a fixed incidence matrix $A$, which corresponds to the all-activated case in the definition of $\boldsymbol{A}(\theta, x)$ above, and which maps input neurons to the path they belong to:

$$
A_{p,u} := \begin{cases} 1 & \text{if } u \in N_{\text{in}} \text{ and } p_0 = u, \\ 1 & \text{if } u = v_{\texttt{bias}} \text{ and } p_0 \notin N_{\text{in}}, \\ 0 & \text{otherwise,} \end{cases}
$$

so $A \in \{0, 1\}^{\mathcal{P} \times (|N_{\text{in}}| + 1)}$ depends *only* on the graph.

### A.4. Key inner-product identity

With our new notations, Equation (4) (corresponding to Theorem A.1 in (Gonon et al., 2024a)) can be rewritten as:

$$
R_\theta(x) = \Big\langle \underbrace{\Phi(\theta) \odot a(\theta, x)}_{\text{path weights}}, \ \underbrace{A}_{\text{fixed incidence}} \begin{pmatrix} x \\ 1 \end{pmatrix} \Big\rangle. \tag{4$'$}
$$

## B. Proof of Theorem 4.1

In this section, we prove a slightly stronger version of Theorem 4.1. We do not state this stronger version in the main body as it requires having in mind the definition of the path-lifting $\Phi$, recalled in Definition A.5, to understand the following notations. For parameters $\theta$, we will denote $\Phi^I(\theta)$ (resp. $\Phi^H(\theta)$) the sub-vector of $\Phi(\theta)$ corresponding to the coordinates associated with paths starting from an input (resp. hidden) neuron. Thus, $\Phi(\theta)$ is the concatenation of $\Phi^I(\theta)$ and $\Phi^H(\theta)$.

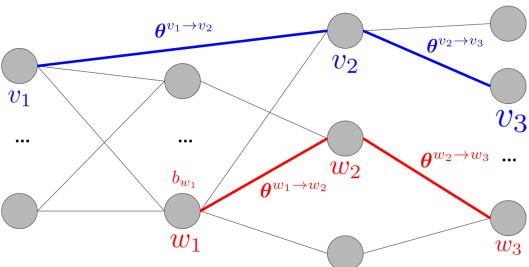

$$A \quad = \quad \begin{array}{c} \mathcal{P}_I \Big\{ \ p \\ \\ \mathcal{P}_H \Big\{ \ p' \end{array} \left( \begin{array}{c} \overbrace{\qquad\qquad}^{N_{\mathrm{in}}} \quad v_{\mathtt{bias}} \\ v_1 \\ \cdots \\ 0 \ \ldots \ 0 \ \ 1 \ \ 0 \ \ldots \ 0 \ \vdots \ 0 \\ \cdots \\ \text{-----------} \\ \vdots \\ 0 \qquad\qquad\qquad 1 \\ \vdots \end{array} \right)$$

Figure 5: The coordinate of the path-lifting $\Phi$ associated with the path $p = v_1 \to v_2 \to v_3$ is $\Phi_p(\theta) = \theta^{v_1 \to v_2} \theta^{v_2 \to v_3}$ since it starts from an input neuron (Definition A.5). While the path $p' = w_1 \to w_2 \to w_3$ starts from a hidden neuron (in $N \setminus (N_{\mathrm{in}} \cup N_{\mathrm{out}})$), so there is also the bias of $w_1$ to take into account: $\Phi_{p'}(\theta) = b_{w_1} \theta^{w_1 \to w_2} \theta^{w_2 \to w_3}$. As specified in Definition A.6, the columns of the incidence matrix $A$ are indexed by $N_{\mathrm{in}} \cup \{v_{\mathtt{bias}}\}$ and its rows are indexed by $\mathcal{P} = \mathcal{P}_I \cup \mathcal{P}_H$, with $\mathcal{P}_I$ the set of paths in $\mathcal{P}$ starting from an input neuron, and $\mathcal{P}_H$ the set of paths starting from a hidden neuron.

**Theorem B.1.** *Consider a ReLU neural network as in Definition A.2, with output dimension equal to one. Consider associated parameters $\theta, \theta'$. If for every coordinate $i$, $\theta_i$ and $\theta'_i$ have the same signs or at least one of them is zero ($\theta_i \theta'_i \geqslant 0$), we have for every input $x$:*

$$|R_\theta(x) - R_{\theta'}(x)| \leqslant \|x\|_\infty \|\Phi^I(\theta) - \Phi^I(\theta')\|_1 + \|\Phi^H(\theta) - \Phi^H(\theta')\|_1. \tag{15}$$

*Moreover, for every neural network architecture, there are non-negative parameters $\theta \neq \theta'$ and a non-negative input $x$ such that Inequality (5) is an equality.*

Theorem B.1 is intentionally stated with scalar output to avoid imposing a specific norm on the outputs; readers can naturally extend it to the vector-valued setting using the norm most relevant to their application. As an example, we derive the next corollary for $\ell^q$-norms on the outputs, which corresponds to the Theorem 4.1 given in the text body (except for the equality case, which is also an easy consequence of the equality case of Inequality (15)).

**Corollary B.2.** *Consider an exponent $q \in [1, \infty)$ and a ReLU neural network as in Definition A.2. Consider associated parameters $\theta, \theta'$. If for every coordinate $i$, it holds $\theta_i \theta'_i \geqslant 0$, then for every input $x \in \mathbb{R}^{d_{in}}$:*

$$\|R_\theta(x) - R_{\theta'}(x)\|_q \leqslant \max(\|x\|_\infty, 1) \|\Phi(\theta) - \Phi(\theta')\|_1.$$

*Proof of Corollary B.2.* By definition of the model, it holds:

$$\|R_\theta(x) - R_{\theta'}(x)\|_q^q = \sum_{v \in N_{out}} |v(\theta, x) - v(\theta', x)|^q.$$

Recall that $\Phi^{\to v}$ is the path-lifting associated with the sub-graph $G^{\to v}$ (Definition A.5). By Theorem B.1, it holds:

$$|v(\theta, x) - v(\theta', x)|^q \leqslant \max(\|x\|_\infty^q, 1) \|\Phi^{\to v}(\theta) - \Phi^{\to v}(\theta')\|_1^q.$$

Since $\Phi(\theta) = (\Phi^{\to v}(\theta))_{v \in N_{out}}$, this implies:

$$\|R_\theta(x) - R_{\theta'}(x)\|_q^q \leqslant \max(\|x\|_\infty^q, 1) \|\Phi(\theta) - \Phi(\theta')\|_1^q. \qquad \square$$

Figure 6: Counter-example showing that the conclusion of Theorem 4.1 does not hold when the parameters have opposite signs. If the hidden neurons are ReLU neurons, the left network implements $R_\theta(x) = \text{ReLU}(x)$ (with $\theta = (1 \quad 1)^T$) and the right network implements $R_{\theta'}(x) = -\text{ReLU}(-x)$ (with $\theta' = (-1 \quad -1)^T$). Inequality (5) does not hold since there is a single path and the product of the weights along this path is equal to one in both cases, so that $\Phi(\theta) = \Phi(\theta') = 1$ (cf Section 3) while these two functions are nonzero and have disjoint supports.

*Proof of Theorem B.1.* A geometric illustration of the spirit of the proof is given in Figure 3, as detailed in the figure caption.

**Step 1 – Reduction to non-zero coordinates.** Since both sides of (15) are continuous in $(\theta, \theta')$, without loss of generality it is enough to prove it for weight vectors $\theta, \theta'$ with no zero entries.

**Step 2 – Proof for parameters leading to the same activations $a(\theta, x) = a(\theta', x)$.** If the two parameters activate exactly the same paths on $x$, then $(4')$ yields

$$|R_\theta(x) - R_{\theta'}(x)| = \left|\left\langle (\Phi(\theta) - \Phi(\theta')) \odot a(\theta, x), \ A \begin{pmatrix} x \\ 1 \end{pmatrix} \right\rangle\right| \leqslant \|(\Phi(\theta) - \Phi(\theta')) \odot a(\theta, x)\|_1 \left\| A \begin{pmatrix} x \\ 1 \end{pmatrix} \right\|_\infty.$$

Because $A$ is binary with at most one "1" per row, $\left\| A \begin{pmatrix} x \\ 1 \end{pmatrix} \right\|_1 \leqslant \left\| \begin{pmatrix} x \\ 1 \end{pmatrix} \right\|_1$. Moreover, $a(\theta, x)$ is a binary vector so $\|(\Phi(\theta) - \Phi(\theta')) \odot a(\theta, x)\|_1 \leqslant \|\Phi(\theta) - \Phi(\theta')\|_1$. This gives the bound of Theorem 4.1 in the simple case where $a(\theta, x) = a(\theta', x)$:

$$|R_\theta(x) - R_{\theta'}(x)| \leqslant \max(\|x\|_\infty, 1) \|\Phi(\theta) - \Phi(\theta')\|_1.$$

To prove the slightly stronger bound appearing in Theorem B.1, first split the paths depending on whether they start at an input neuron or at a hidden neuron:

$$\left|\langle (\Phi(\theta) - \Phi(\theta')) \odot a(\theta, x), \ A \begin{pmatrix} x \\ 1 \end{pmatrix} \rangle\right| \leqslant \left|\langle (\Phi^I(\theta) - \Phi^I(\theta')) \odot a^I(\theta, x), \ A^I x \rangle\right| + \left|\langle (\Phi^H(\theta) - \Phi^H(\theta')) \odot a^H(\theta, x), \ A^H \rangle\right|$$

and then apply the same argument on each part to get:

$$|R_\theta(x) - R_{\theta'}(x)| \leqslant \|x\|_\infty \|\Phi^I(\theta) - \Phi^I(\theta')\|_1 + \|\Phi^H(\theta) - \Phi^H(\theta')\|_1.$$

**Step 3 – A bound for a trajectory with finitely many break-points.** Let $t \mapsto \theta(t)$ be any continuous curve from $\theta$ to $\theta'$ such that *the activation vector $a(\theta(t), x)$ is constant on finitely many intervals* $(t_k, t_{k+1})$ with $t_k < t_{k+1}$ and $[0,1] = \cup_{k=0}^m [t_k, t_{k+1}]$. Applying Step 2 on every interval[6], and summing gives

$$|R_\theta(x) - R_{\theta'}(x)| \ \leqslant \ \|x\|_\infty \sum_k \|\Phi^I(\theta(t_{k+1})) - \Phi^I(\theta(t_k))\|_1 \ + \ \sum_k \|\Phi^H(\theta(t_{k+1})) - \Phi^H(\theta(t_k))\|_1. \tag{A.2}$$

**Step 4 – Construction of a monotone path in log-space.** For each coordinate index $i$ of the vector $\theta$ define

$$\theta_i(t) \ = \ \mathrm{sgn}(\theta_i) \, |\theta_i|^{1-t} \, |\theta_i'|^t, \qquad t \in [0,1]. \tag{A.3}$$

This trajectory $t \to \theta(t)$ is well-defined since by Step 1 we assumed without loss of generality that the coordinates of $\theta$ and $\theta'$ are non-zero. Moreover, since $\mathrm{sgn}(\theta) = \mathrm{sgn}(\theta')$, this trajectory goes from $\theta$ to $\theta'$ and we can use (A.2) provided that this path has only finitely many break-points.

For every path $p$, the scalar function $t \mapsto \Phi_p(\theta(t)) = |\Phi_p(\theta)|^{1-t} |\Phi_p(\theta')|^t$ is monotone, so developing the $\ell^1$-norms in (A.2) yields sums that telescope exactly:

$$\sum_k \|\Phi^I(\theta_{k+1}) - \Phi^I(\theta_k)\|_1 = \|\Phi^I(\theta) - \Phi^I(\theta')\|_1, \quad \text{and similarly for } \Phi^H.$$

Thus Theorem B.1 follows from (A.2) *provided* the path has only finitely many break-points.

**Step 5 – Proving the existence of finitely many break-points (technical).** Each coordinate in (A.3) is an analytic function of $t$, and the activation of a ReLU or max-pool neuron evaluated on an analytic input can only change at isolated roots. On the compact interval $[0,1]$ there can be only finitely many such roots. Lemma B.3 formalizes this argument and completes the proof of (15).

To prove Theorem B.1, it remains to prove the claim about the equality case: we must find $\theta \neq \theta'$ and an input $x$ such that the inequality is actually an equality.

**Sharpness of the bound (equality cases) in Theorem B.1** Consider an input neuron $v_0$ and a path $p = v_0 \to v_1 \to \cdots \to v_d$. Define two parameter vectors that differ only on that path:

$$\theta_{v_\ell \to v_{\ell+1}} = a > 0, \qquad \theta'_{v_\ell \to v_{\ell+1}} = b > 0, \quad \ell = 0, \ldots, d-1,$$

and set every other coordinate of $\theta, \theta'$ to 0.

Choose the input $x$ with $x_{v_0} > 0$ and all other coordinates equal to 0. Because the signal propagates solely along $p$,

$$R_\theta(x) = a^d \, x_{v_0}, \qquad R_{\theta'}(x) = b^d \, x_{v_0}.$$

For the path–lifting, only the coordinate $\Phi_p$ changes, hence

$$\|\Phi^I(\theta) - \Phi^I(\theta')\|_1 = |a^d - b^d|, \qquad \|\Phi^H(\theta) - \Phi^H(\theta')\|_1 = 0.$$

Since $\|x\|_\infty = x_{v_0}$, inequality (15) is an equality:

$$|a^d - b^d| \, x_{v_0} \ = \ \|x\|_\infty \, \|\Phi^I(\theta) - \Phi^I(\theta')\|_1 \ + \ \|\Phi^H(\theta) - \Phi^H(\theta')\|_1,$$

Thus the bound of Theorem B.1 cannot be improved in general. $\qquad \square$

---

[6]Formally, apply Step 2 to a pair $\theta(t), \theta(t')$, with $t, t'$ in the *open* interval $(t_k, t_{k+1})$, let $t \to t_k$ and $t' \to t_{k+1}$, and conclude by continuity of both sides.

**Lemma B.3.** *Fix $n \in \mathbb{N}$ inputs $x_1, \ldots, x_n \in \mathbb{R}^{d_{in}}$ and two parameter vectors $\theta, \theta'$ with no zero coordinates. Let $\theta(t)$ be the geometric trajectory* (A.3). *There are finitely many points $0 = t_0 < t_1 < \cdots < t_m = 1$ such that for every input $x_i$ the path-activation vector $a(\theta(t), x_i)$ is constant on each open interval $(t_k, t_{k+1})$.*

*Proof of Lemma B.3.* **Step 1 – reduce to a single input.** If a finite breakpoint set works for each $x_i$ individually, their union works for all inputs. We therefore fix one arbitrary input $x$.

**Step 2 – property to prove for each neuron.** For a neuron $v$ define

$$
\mathbf{P}(v) : \begin{cases} \text{there exist finitely many breakpoints } 0 = t_0 < t_1 < \cdots < t_m = 1 \text{ such that} \\ \quad t \mapsto v(\theta(t), x) \text{ is analytic on every } [t_k, t_{k+1}], \\ \quad t \mapsto a_v(\theta(t), x) \text{ and } t \mapsto a_{u \to v}(\theta(t), x) \; \forall u \in \operatorname{ant}(v) \\ \quad \text{are constant on } (t_k, t_{k+1}). \end{cases}
$$

If $\mathbf{P}(v)$ holds for every neuron $v$, the union of their breakpoints gives finitely many intervals on which *all* edge and path activations are frozen, completing the lemma.

**Step 3 – prove $\mathbf{P}(v)$ by topological induction.**

We perform induction on a topological sorting (Cormen et al., 2009, Section 22.4) of the underlying DAG. We start with input neurons $v$ since by Definition A.2, these are the ones without antecedents so they are the first to appear in a topological sorting.

**Initialization: Input neurons.** $v(\theta, x) = x_v$ does not depend on $\theta$, hence $a_v(\cdot, x) \equiv 1$; $\mathbf{P}(v)$ holds with $m = 1$.

**Induction:** Now consider a neuron $v \notin N_{\text{in}}$ and assume $\mathbf{P}(u)$ to hold for every neuron $u$ coming before $v$ in the topoligical sorting. There are finitely many breakpoints $0 = t_0 < t_1 < \cdots < t_m = 1$ such that for every $u \in \operatorname{ant}(v)$ and every $k$, the map $t \in [t_k, t_{k+1}] \mapsto u(\theta(t), x)$ is analytic. We distinguish three cases depending on the activation function of the neuron $v$.

*(i) Identity neuron.* $v(\theta(t), x) = b_v + \sum_{u \in \operatorname{ant}(v)} u(\theta(t), x)\theta_{u \to v}(t)$ is analytic on the same intervals $[t_k, t_{k+1}]$ because each factor is analytic; $a_v \equiv 1$. Thus $\mathbf{P}(v)$ inherits the finite breakpoint set of its antecedents.

*(ii) ReLU neuron.* The pre-activation $\operatorname{pre}_v(t) := b_v + \sum_{u \in \operatorname{ant}(v)} u(\theta(t), x)\theta_{u \to v}(t)$ is analytic on each $[t_k, t_{k+1}]$ by induction. Either $\operatorname{pre}_v$ is identically zero, in which case $a_v \equiv 0$, or its zero set is finite (as an analytic function on a compact domain), so the sign of $\operatorname{pre}_v$ (and therefore $a_v$ and each $a_{u \to v}$) is constant between consecutive zeros. Hence $\mathbf{P}(v)$ holds.

*(iii) $K$-max-pool neuron.* The output of $v$ is the $K$-th largest component of $\operatorname{pre}_v := \left( u(\theta(t), x)\theta_{u \to v}(t) \right)_{u \in \operatorname{ant}(v)}$. Each coordinate of $\operatorname{pre}_v$ is analytic on each $[t_k, t_{k+1}]$ by induction. Two coordinates can swap order only at isolated $t$ where their analytic difference becomes zero, so the ranking—and thus the selected $K$-th value—changes only finitely many times. Thus $\mathbf{P}(v)$ holds.

By topological induction $\mathbf{P}(v)$ is true for every neuron. The argument in Step 2 then gives the desired global breakpoint set. $\qquad\square$

## C. Proof of Lemma 5.2

Rescaling-invariance is a direct consequence of the known properties of the path-lifting $\Phi$ (Gonon et al., 2024a).

In the case of a singleton $I = \{i\}$, as already evoked, (13) simply follows from (5) and the definition of `Path-Mag`. When $|I| \geqslant 2$, consider any enumeration $i_j$, $1 \leqslant j \leqslant |I|$ of elements in $I$, and $s_j := \mathbf{1}_G - \sum_{\ell=1}^{j} e_{i_\ell} = \mathbf{1}_G - \mathbf{1}_{\cup_{\ell=1}^{j} \{i_\ell\}}$ (as well as $s_0 := \mathbf{1}_G$): since the pair $(\theta, s \odot \theta)$ –as well as the pairs $(s_{j-1} \odot \theta, s_j \odot \theta)$– satisfies the assumptions of Lemma 4.2, and

$s_j \odot s_{j-1} = s_j$ we have

$$
\begin{aligned}
\|\Phi(\theta) - \Phi(s \odot \theta)\|_1 &\underset{(7)}{=} \|\Phi(\theta)\|_1 - \|\Phi(s \odot \theta)\|_1 \\
&= \sum_{j=1}^{|I|} \|\Phi(s_{j-1} \odot \theta)\|_1 - \|\Phi(s_j \odot \theta)\|_1 \\
&\underset{(7)}{=} \sum_{j=1}^{|I|} \|\Phi(s_{j-1} \odot \theta) - \Phi(s_j \odot (s_{j-1} \odot \theta))\|_1 \\
&\underset{(10)}{=} \sum_{j=1}^{|I|} \texttt{Path-Mag}(s_{j-1} \odot \theta, i_j) \\
&\underset{(12)}{\leqslant} \sum_{j=1}^{|I|} \texttt{Path-Mag}(\theta, i_j).
\end{aligned}
$$

Finally, to establish (14), observe that for each path $p$ we have $|\Phi_p(\theta)| = \prod_{j \in p} |\theta_j|$ so, for each $i \in p$ (NB: $i$ can index either an edge in the path or the first neuron of $p$ when $p$ starts from a hidden or output neuron, in which case $\theta_i$ is the associated bias) it holds

$$
\frac{\partial}{\partial \theta_i} |\Phi_p(\theta)| = \mathrm{sgn}(\theta_i) \prod_{j \in p, j \neq i} |\theta_j|.
$$

Because $\mathrm{sgn}(\theta_i)\theta_i = |\theta_i|$ we get that, when $i \in p$,

$$
\theta_i \cdot \frac{\partial}{\partial \theta_i} |\Phi_p(\theta)| = |\Phi_p(\theta)|.
$$

Summing over all paths for a given index $i$ shows that

$$
\begin{aligned}
(\theta \odot \nabla_\theta \|\Phi(\theta)\|_1)_i &= \theta_i \frac{\partial}{\partial \theta_i} \sum_{p \in \mathcal{P}} |\Phi_p(\theta)| \\
&= \theta_i \sum_{p \in \mathcal{P}: i \in p} \frac{\partial}{\partial \theta_i} |\Phi_p(\theta)| \\
&= \sum_{p \in \mathcal{P}: i \in p} |\Phi_p(\theta)| \\
&\underset{(12)}{=} \texttt{Path-Mag}(\theta, i).
\end{aligned}
$$

## D. Proof-of-concept: accuracy of path-magnitude pruning

To provide a proof-of-concept of the utility of the main Lipschitz bound in Theorem 4.1 for pruning, we implement the following "prune and finetune" procedure:

1. **train**: we train a dense network,

2. **rescale (optional)**: we apply a random rescale to the trained weights (this includes biases),

3. **prune**: we prune the resulting network,

4. **rewind**: we rewind the weights to their value after a few initial epochs (standard in the lottery ticket literature to enhance performance (Frankle et al., 2020)),

5. **finetune**: we retrain the pruned network, with the pruned weights frozen to zero and the other ones initialized from their rewinded values.

Doing that to prune $p = 40\%$ of the weights at once of a ResNet18 trained on ImageNet-1k, we observe that (Figure 4):

- **without random rescale** (plain lines), the test accuracy obtained at the end is *similar* for both magnitude pruning and path-magnitude pruning;

- **with random rescale** (dotted lines – the one associated with path-magnitude pruning is invisible as it coincides with the corresponding plain line), magnitude pruning suffers a large drop of top-1 test accuracy, which is not the case of path-magnitude pruning since it makes the process invariant to potential rescaling.

We observe similar results when pruning between $p = 10\%$ and $p = 80\%$ of the weights at once, see Table 5.

Table 5: Extended version of Table 4. Top-1 accuracy after pruning, optional rescale, rewind and retrain, as a function of the pruning level. $(*)$ = results valid with as well as without rescaling, as path-magnitude pruning is invariant to rescaling.

| Pruning level | none | 10% | 20% | 40% | 60% | 80% |
|---|---|---|---|---|---|---|
| Path-Magnitude $(*)$ | | 68.6 | 68.8 | 68.6 | 67.9 | 66.0 |
| Magnitude w/o Random Rescale | 67.7% | 69.0 | 69.0 | 68.8 | 68.2 | 66.5 |
| Magnitude w/ Random Rescale | | 68.8 | 68.7 | 63.1 | 57.5 | 15.8 |

We now give details on each stage of the procedure.

**1. Train.** We train a dense ResNet18 (He et al., 2016) on ImageNet-1k, using $99\%$ of the 1,281,167 images of the training set for training, the other $1\%$ for validation. We use SGD for 90 epochs, learning rate 0.1, weight-decay 0.0001, batch size 1024, classical ImageNet data normalization, and a multi-step scheduler where the learning rate is divided by 10 at epochs 30, 60 and 80. The epoch out of the 90 ones with maximum validation top-1 accuracy is considered as the final epoch. Doing 90 epochs took us about 18 hours on a single A100-40GB GPU.

**2. Random rescaling.** Consider a pair of consecutive convolutional layers in the same basic block of the ResNet18 architecture, for instance the ones of the first basic block: `model.layer1[0].conv1` and `model.layer1[0].conv2` in PyTorch, with `model` being the ResNet18. Denote by $C$ the number of output channels of the first convolutional layer, which is also the number of input channels of the second one. For each channel $c \in [\![1, C]\!]$, we choose uniformly at random a rescaling factor $\lambda \in \{1, 128, 4096\}$ and multiply the output channel $c$ of the first convolutional layer by $\lambda$, and divide the input channel $c$ of the second convolutional layer by $\lambda$. In order to preserve the input-output relationship, we also multiply by $\lambda$ the running mean and the bias of the batch normalization layer that is in between (`model.layer1[0].bn1` in the previous example). Here is an illustrative Python code (that should be applied to the correct layer weights as described above):

```
factors = np.array([1, 128, 4096])

out_channels1, _, _, _ = weights_conv1.shape

for out in range(out_channels1):
            factor = np.random.choice(factors)
            weights_conv1[out, :, :, :] *= factor
            weights_conv2[:, out, :, :] /= factor
            running_mean[out] *= factor
            bias[out] *= factor
```

**3. Pruning.** At the end of the training phase, we globally prune (i.e. set to zero) $p\%$ of the remaining weights in all the convolutional layers plus the final fully connected layer.

**4. Rewinding.** We save the mask and rewind the weights to their values after the first 5 epochs of the dense network, and train for 85 remaining epochs. This exactly corresponds to the hyperparameters and pruning algorithm of the lottery ticket literature (Frankle et al., 2021).

**5. Finetune.** This is done in the same conditions as the training phase.

# E. Computational cost: comparing pruning criteria

This section details how the results of Table 3 were obtained.

## E.1. Hardware and software

All experiments were performed on an NVIDIA A100-PCIE-40GB GPU, with CPU Intel(R) Xeon(R) Silver 4215R CPU @ 3.20GHz. We used `PyTorch` (version *2.2*, with `CUDA` *12.1* and `cuDNN` *8.9* enabled) to implement model loading, inference, and custom pruning-cost computation. All timings were taken using the `torch.utils.benchmark` module, synchronizing the GPU to ensure accurate measurement of wall-clock time.

## E.2. Benchmarked code

**Single-forward pass.** We fed a tensor `torch.randn(B, 3, 224, 224)` to each model (batch size $B = 1$ or $B = 128$, $224 \times 224$ RGB image).

**Path-magnitude scores.** We followed the recipe given in (14): $(\texttt{Path-Mag}(\theta, i))_i = \theta \odot \nabla_\theta \|\Phi(\theta)\|_1$. To do that, we computed the path-norm $\|\Phi(\theta)\|_1$ using the function `get_path_norm` we released online at `github.com/agonon/pathnorm_toolkit` (Gonon et al., 2024b). And we simply added one line to auto-differentiate the computations and multiply the result pointwise with the parameters $\theta$. Thus, our code (see (Gonon et al., 2025) and `github.com/agonon/pathnorm_toolkit` for updates) has the following structure:

- it starts by replacing max-pooling *neurons* by summation *neurons*, or equivalently max-pooling *layers* by convolutional *layers* (following the recipe given in (Gonon et al., 2024a) to compute correctly the path-norm),

- it replaces each weight by its absolute value,

- it does a forward pass to compute the path-norm,

- here we added auto-differentiation (backwarding the path-norm computations), and pointwise multiplication with original weights,

- and it finally reverts to the original maxpool layers and the weights' value to restore the original network.

Table 3 reports the time to do all this.

**Magnitude scores.** It takes as input a torch model, and does a simple loop over all model's parameters:

- to check if these are the parameters of a `torch.nn.Linear` or `torch.nn.Conv2d` module,

- if this is the case, it adds to a list the absolute values of these weights.

**Loss-sensitivity scores (LeCun et al., 1989).** In the Optimal Brain Damage (OBD) framework introduced in (LeCun et al., 1989), each weight $\theta_i$ in the network is assigned a score approximating the expected increase in loss if $\theta_i$ were pruned (set to zero). The score of $\theta_i$ is defined by:

$$\mathrm{OBD}(\theta, i) = \frac{1}{2} h_{ii} \theta_i^2,$$

where $h_{ii}$ is the diagonal entry of the Hessian matrix $H = \nabla^2 \ell$ of the empirical loss

$$\ell(\theta) = \sum_{k=1}^{n} \ell(R_\theta(x_k), y_k)$$

with respect to the parameters $\theta$. As we could not locate a proof of the rescaling-invariance of OBD we give below a short proof, before discussing its numerical computation.

*Rescaling-invariance.* Denote $D = \texttt{diag}(\lambda_i)$ a diagonal rescaling matrix such that for each $\theta$ the parameters $\theta' := D\theta$ are rescaling-equivalent to $\theta$. This implies that $R_\theta(x_k) = R_{D\theta}(x_k)$ for each training sample $x_k$ and every $\theta$, hence

$\ell(\theta) = \ell(D\theta)$ for every $\theta$. Simple calculus then yields equality of the Jacobians $\partial\ell(\theta) = \partial\ell(D\theta)D$, i.e., since $D$ is symmetric, taking the transpose

$$\nabla\ell(\theta) = D\nabla\ell(D\theta), \quad \forall\theta,$$

that is to say $\nabla\ell(\cdot) = D\nabla\ell(D\cdot)$. Differentiating once more yields

$$H(\theta) = \nabla^2\ell(\theta) = \partial[\nabla\ell](\theta) = \partial[D\nabla\ell(D\cdot)](\theta) = D\partial[\nabla\ell(D\cdot)](\theta) = D\partial[\nabla\ell(\cdot)](D\theta)D = DH(D\theta)D.$$

Extracting the $i$-th diagonal entry yields $h_{ii}(\theta) = \lambda_i^2 h_{ii}(D\theta)$ (and more generally $h_{ij}(\theta) = \lambda_i\lambda_j h_{ij}(D\theta)$), hence

$$\mathrm{OBD}(D\theta, i) = \frac{1}{2}h_{ii}(D\theta)((D\theta)_i)^2 = \frac{1}{2}h_{ii}(D\theta)(\lambda_i\theta_i)^2 = \frac{1}{2}[h_{ii}(D\theta)\lambda_i^2]\theta_i^2 = \frac{1}{2}h_{ii}(\theta)\theta_i^2 = \mathrm{OBD}(\theta, i). \quad (16)$$

*Computation.* Computing the full Hessian matrix $H$ exactly would be prohibitive for large networks. Instead, a well-known variant of Hutchinson's trick (Bekas et al., 2007) is that its diagonal can be computed as

$$\mathrm{diag}(H) = \mathbb{E}_v\big[(Hv) \odot v\big]$$

where the expectation is over Rademacher vectors $v$ (i.i.d. uniform $v_i \in \{-1, 1\}$) and where $\odot$ denotes pointwise multiplication. In practice, we approximate it as follows:

- draw a *single vector* $v$ as above,

- compute the Hessian-vector product $Hv$ using the "reverse-over-forward" higher-order autodiff in PyTorch's `torch.func` API,

- deduce the estimate $\mathrm{diag}(H) \simeq (Hv) \odot v =: u$,

- finally estimate $\mathrm{OBD} \simeq \frac{1}{2}u \odot \theta \odot \theta = \frac{1}{2}(Hv) \odot v \odot \theta \odot \theta$.

The performance to do all this depends on the size of the batch on which is computed the loss, as the cost of the Hessian-vector product $Hv$ depends on it. Table 3 reports the milliseconds required for this entire procedure on batch sizes of 1 and 128, listing corresponding values as $x - y$.

This approximation is *not* rescaling-invariant in general. Indeed, we have

$$((Hv) \odot v \odot \theta \odot \theta)_i = (Hv)_i \cdot v_i \cdot \theta_i^2 = \left(\sum_j h_{ij}(\theta)v_j\right)v_i\theta_i^2$$

$$\underset{(16)}{=} \left(\sum_j \lambda_i\lambda_j h_{ij}(D\theta)v_j\right)v_i\theta_i^2 = \left(\sum_j \lambda_j h_{ij}(D\theta)v_j\right)v_i\lambda_i\theta_i^2$$

which would be the same as the estimate made for $D\theta$ if and only if it were equal to

$$\left(\sum_j h_{ij}(D\theta)v_j\right)v_i\lambda_i^2\theta_i^2.$$

There is no reason for this to happen (and it did not happen in any of our experiments). For instance, take $v_i = \theta_i = 1$ for every $i$, we would need $\sum_j \lambda_j h_{ij}(D\theta) = \lambda_i \sum_j h_{ij}(D\theta)$, which is the same as saying that $\lambda$ is an eigenvector of $H(D\theta)$ with eigenvalue 1.

## F. Lipschitz property of $\Phi$: proof of Lemma 4.3

We first establish Lipschitz properties of $\theta \mapsto \Phi(\theta)$. Combined with the main result of this paper, Theorem 4.1, or with Corollary B.2, they establish a Lipschitz property of $\theta \mapsto R_\theta(x)$ for each $x$, and of the functional map $\theta \mapsto R_\theta(\cdot)$ in the uniform norm on any bounded domain. This is complementary to the Lipschitz property of $x \mapsto R_\theta(x)$ studied elsewhere in the literature, see e.g. (Gonon et al., 2024a).

**Lemma F.1.** *Consider $q \in [1, \infty)$, parameters $\theta$ and $\theta'$, and a neuron $v$. Then, it holds:*

$$\|\Phi^{\to v}(\theta) - \Phi^{\to v}(\theta')\|_q^q$$
$$\leqslant \max_{p \in \mathcal{P}^{\to v}} \sum_{\ell=1}^{\text{length}(p)} \left( \prod_{k=\ell+1}^{\text{length}(p)} \|\theta^{\to p_k}\|_q^q \right) \left( |b_{p_\ell} - b'_{p_\ell}|^q + \|\theta^{\to p_\ell} - (\theta')^{\to p_\ell}\|_q^q \max_{u \in \text{ant}(p_\ell)} \|\Phi^{\to u}(\theta')\|_q^q \right) \tag{17}$$

*with the convention that an empty sum and product are respectively equal to zero and one.*

Note that when all the paths in $\mathcal{P}^{\to v}$ have the same length $L$, Inequality (17) is homogeneous: multiplying both $\theta$ and $\theta'$ coordinate-wise by a scalar $\lambda$ scales both sides of the equations by $\lambda^L$.

*Proof.* The proof of Inequality (17) goes by induction on a topological sorting of the graph. The first neurons of the sorting are the neurons without antecedents, *i.e.*, the input neurons by definition. Consider an input neuron $v$. There is only a single path ending at $v$: the path $p = v$. By Definition A.5, $\Phi^{\to v}(\cdot) = \Phi_v(\cdot) = 1$ so the left hand-side is zero. On the right-hand side, there is only a single choice for a path ending at $v$: this is the path $p = v$ that starts and ends at $v$. Thus $D = 0$, and the maximum is zero (empty sum). This proves Inequality (17) for input neurons.

Consider a neuron $v \notin N_{\text{in}}$ and assume that this is true for every neuron before $v$ in the considered topological sorting. Recall that, by definition, $\Phi^{\to v}$ is the path-lifting of $G^{\to v}$ (see Definition A.5). The paths in $G^{\to v}$ are $p = v$, and the paths going through antecedents of $v$ ($v$ has antecedents since it is not an input neuron). So we have $\Phi^{\to v}(\theta) = \begin{pmatrix} (\Phi^{\to u} \times \theta^{u \to v})_{u \in \text{ant}(v)} \\ b_v \end{pmatrix}$, where we again recall that $\Phi^{\to u}(\cdot) = 1$ for input neurons $u$, and $b_u = 0$ for $*$-max-pooling neurons. Thus, we have:

$$\|\Phi^{\to v}(\theta) - \Phi^{\to v}(\theta')\|_q^q$$
$$= |b_v - b'_v|^q + \sum_{u \in \text{ant}(v)} \|\Phi^{\to u}(\theta) \times \theta^{u \to v} - \Phi^{\to u}(\theta') \times (\theta')^{u \to v}\|_q^q$$
$$\leqslant |b_v - b'_v|^q + \sum_{u \in \text{ant}(v)} \left( \|\Phi^{\to u}(\theta) - \Phi^{\to u}(\theta')\|_q^q |\theta^{u \to v}|^q + \|\Phi^{\to u}(\theta')\|_q^q |\theta^{u \to v} - (\theta')^{u \to v}|^q \right)$$
$$\leqslant |b_v - b'_v|^q + \|\theta^{\to v}\|_q^q \max_{u \in \text{ant}(v)} \|\Phi^{\to u}(\theta) - \Phi^{\to u}(\theta')\|_q^q + \|\theta^{\to v} - (\theta')^{\to v}\|_q^q \max_{u \in \text{ant}(v)} \|\Phi^{\to u}(\theta')\|_q^q.$$

Using the induction hypothesis (Inequality (17)) on the antecedents of $v$ and observing that $p \in \mathcal{P}^{\to v}$ if, and only if there are $u \in \text{ant}(v), r \in \mathcal{P}^{\to u}$ such that $p = r \to v$ gives (we highlight in **blue** the important changes):

$$\|\Phi^{\to v}(\theta) - \Phi^{\to v}(\theta)\|_q^q \leqslant |b_v - b'_v|^q + \|\theta^{\to v} - (\theta')^{\to v}\|_q^q \max_{u \in \text{ant}(v)} \|\Phi^{\to u}(\theta')\|_q^q$$

$$+ \|\boldsymbol{\theta^{\to v}}\|_q^q \max_{\boldsymbol{u \in \text{ant}(v)}} \max_{\boldsymbol{r \in \mathcal{P}^{\to u}}} \sum_{\ell=1}^{\text{length}(r)} \left( \prod_{k=\ell+1}^{\text{length}(r)} \|\theta^{\to r_k}\|_q^q \right) \left( |b_{r_\ell} - b'_{r_\ell}|^q + \|\theta^{\to r_\ell} - (\theta')^{\to r_\ell}\|_q^q \max_{w \in \text{ant}(r_\ell)} \|\Phi^{\to w}(\theta')\|_q^q \right).$$

$$= |b_v - b'_v|^q + \|\theta^{\to v} - (\theta')^{\to v}\|_q^q \max_{u \in \text{ant}(v)} \|\Phi^{\to u}(\theta')\|_q^q$$

$$+ \max_{\boldsymbol{p \in \mathcal{P}^{\to v}}} \sum_{\ell=1}^{\text{length}(\boldsymbol{p})-1} \left( \prod_{k=\ell+1}^{\text{length}(p)} \|\theta^{\to p_k}\|_q^q \right) \left( |b_{p_\ell} - b'_{p_\ell}|^q + \|\theta^{\to p_\ell} - (\theta')^{\to p_\ell}\|_q^q \max_{w \in \text{ant}(p_\ell)} \|\Phi^{\to w}(\theta')\|_q^q \right)$$

$$= \max_{p \in \mathcal{P}^{\to v}} \sum_{\ell=1}^{\text{length}(\boldsymbol{p})} \left( \prod_{k=\ell+1}^{\text{length}(p)} \|\theta^{\to p_k}\|_q^q \right) \left( |b_{p_\ell} - b'_{p_\ell}|^q + \|\theta^{\to p_\ell} - (\theta')^{\to p_\ell}\|_q^q \max_{w \in \text{ant}(p_\ell)} \|\Phi^{\to w}(\theta')\|_q^q \right).$$

This proves Inequality (17) for $v$ and concludes the induction. $\qquad\square$

In the sequel it will be useful to restrict the analysis to *normalized* parameters, defined as parameters $\tilde{\theta}$ such that $\left\| \begin{pmatrix} \tilde{\theta}^{\to v} \\ \tilde{b}_v \end{pmatrix} \right\|_1 \in \{0, 1\}$ for every $v \in N \setminus (N_{\text{out}} \cup N_{\text{in}})$. Thanks to the rescaling-invariance of ReLU neural network

parameterizations, Algorithm 1 in Gonon et al. (2024a) allows to rescale *any* parameters $\theta$ into a normalized version $\tilde{\theta}$ such that $R_{\tilde{\theta}} = R_\theta$ and $\Phi(\theta) = \Phi(\tilde{\theta})$ (Gonon et al., 2024a, Lemma B.2). This implies the next simpler results for normalized parameters.

**Theorem F.2.** *Consider $q \in [1, \infty)$. For every normalized parameters $\theta, \theta'$ obtained as the output of Algorithm 1 in Gonon et al. (2024a), it holds:*

$$\|\Phi(\theta) - \Phi(\theta')\|_q^q \leqslant \sum_{v \in N_{out} \setminus N_{in}} |b_v - b_v'|^q + \|\theta^{\to v} - (\theta')^{\to v}\|_q^q$$

$$+ \min\left(\|\Phi(\theta)\|_q^q, \|\Phi(\theta')\|_q^q\right) \max_{p \in \mathcal{P}: p_{\text{end}} \notin N_{in}} \sum_{\ell=1}^{\texttt{length}(p)-1} \left(|b_{p_\ell} - b_{p_\ell}'|^q + \|\theta^{\to p_\ell} - (\theta')^{\to p_\ell}\|_q^q\right). \quad (18)$$

Denote by $\texttt{N}(\theta)$ the normalized version of $\theta$, obtained as the output of Algorithm 1 in Gonon et al. (2024a). It can be checked that if $\theta = \texttt{N}(\tilde{\theta})$ and $\theta' = \texttt{N}(\tilde{\theta}')$, and if all the paths have the same lengths $L$, then multiplying both $\tilde{\theta}$ and $\tilde{\theta}'$ coordinate-wise by a scalar $\lambda$ does not change their normalized versions $\theta$ and $\theta'$, except for the biases and the incoming weights of all output neurons that are scaled by $\lambda^L$. As a consequence, Inequality (18) is homogeneous: both path-liftings on the left-hand-side and the right-hand-side are multiplied by $\lambda^L$, and so is the sum over $v \in N_{\text{out}} \setminus N_{\text{in}}$ in the right-hand-side, while the maximum over $p$ is unchanged since it only involves normalized coordinates that do not change.

For networks used in practice, it holds $N_{\text{out}} \cap N_{\text{in}} = \emptyset$ so that $N_{\text{out}} \setminus N_{\text{in}}$ is just $N_{\text{out}}$, but the above theorem also covers the somewhat pathological case of DAG architectures $G$ where one or more input neurons are also output neurons.

*Proof of Theorem F.2.* Since $\Phi(\theta) = (\Phi^{\to v}(\theta))_{v \in N_{\text{out}}}$, it holds

$$\|\Phi(\theta) - \Phi(\theta')\|_q^q = \sum_{v \in N_{\text{out}}} \|\Phi^{\to v}(\theta) - \Phi^{\to v}(\theta')\|_q^q.$$

By Definition A.5, it holds for every input neuron $v$: $\Phi^{\to v}(\cdot) = 1$. Thus, the sum can be taken over $v \in N_{\text{out}} \setminus N_{\text{in}}$:

$$\|\Phi(\theta) - \Phi(\theta')\|_q^q = \sum_{v \in N_{\text{out}} \setminus N_{\text{in}}} \|\Phi^{\to v}(\theta) - \Phi^{\to v}(\theta')\|_q^q.$$

Besides, observe that many norms appearing in Inequality (17) are at most one for normalized parameters. Indeed, for such parameters it holds for every $u \in N \setminus (N_{\text{in}} \cup N_{\text{out}})$: $\|\theta^{\to u}\|_q^q \leqslant 1$ (Gonon et al., 2024a, Lemma B.2). As a consequence, for $p \in \mathcal{P}$ and any $\ell \in [\![0, \texttt{length}(p) - 1]\!]$ we have:

$$\prod_{k=\ell+1}^{\texttt{length}(p)} \|\theta^{\to p_k}\|_q^q = \left(\prod_{k=\ell+1}^{\texttt{length}(p)-1} \underbrace{\|\theta^{\to p_k}\|_q^q}_{\leqslant 1}\right) \|\theta^{\to p_{\text{end}}}\|_q^q \leqslant \|\theta^{\to p_{\text{end}}}\|_q^q.$$

Moreover, for normalized parameters $\theta$ and $u \notin N_{\text{out}}$, it also holds $\|\Phi^{\to u}(\theta)\|_q^q \leqslant 1$ (Gonon et al., 2024a, Lemma B.3). Thus, Inequality (17) implies for any $v \in N_{\text{out}}$, and any normalized parameters $\theta$ and $\theta'$:

$$\|\Phi^{\to v}(\theta) - \Phi^{\to v}(\theta')\|_q^q$$

$$\leqslant |b_v - b_v'|^q + \|\theta^{\to v} - (\theta')^{\to v}\|_q^q + \|\theta^{\to v}\|_q^q \max_{p \in \mathcal{P}^{\to v}} \sum_{\ell=1}^{\texttt{length}(p)-1} \left(|b_{p_\ell} - b_{p_\ell}'|^q + \|\theta^{\to p_\ell} - (\theta')^{\to p_\ell}\|_q^q\right).$$

Thus, we get:

$$\|\Phi(\theta) - \Phi(\theta')\|_q^q$$
$$= \sum_{v \in N_{\text{out}} \setminus N_{\text{in}}} \|\Phi^{\to v}(\theta) - \Phi^{\to v}(\theta')\|_q^q$$
$$\leqslant \sum_{v \in N_{\text{out}} \setminus N_{\text{in}}} \left( |b_v - b_v'|^q + \|\theta^{\to v} - (\theta')^{\to v}\|_q^q \right)$$
$$+ \sum_{v \in N_{\text{out}} \setminus N_{\text{in}}} \|\theta^{\to v}\|_q^q \max_{p \in \mathcal{P}^{\to v}} \sum_{\ell=1}^{\texttt{length}(p)-1} \left( |b_{p_\ell} - b_{p_\ell}'|^q + \|\theta^{\to p_\ell} - (\theta')^{\to p_\ell}\|_q^q \right)$$
$$\leqslant \sum_{v \in N_{\text{out}} \setminus N_{\text{in}}} \left( |b_v - b_v'|^q + \|\theta^{\to v} - (\theta')^{\to v}\|_q^q \right)$$
$$+ \left( \sum_{v \in N_{\text{out}} \setminus N_{\text{in}}} \|\theta^{\to v}\|_q^q \right) \max_{p \in \mathcal{P} : p_{\text{end}} \notin N_{\text{in}}} \sum_{\ell=1}^{\texttt{length}(p)-1} \left( |b_{p_\ell} - b_{p_\ell}'|^q + \|\theta^{\to p_\ell} - (\theta')^{\to p_\ell}\|_q^q \right).$$

It remains to use that $\sum_{v \in N_{\text{out}} \setminus N_{\text{in}}} \|\theta^{\to u_v}\|_q^q \leqslant \|\Phi(\theta)\|_q^q$ for normalized parameters $\theta$ (Gonon et al., 2024a, Theorem B.1, case of equality) to conclude that:

$$\|\Phi(\theta) - \Phi(\theta')\|_q^q \leqslant \sum_{v \in N_{\text{out}} \setminus N_{\text{in}}} \left( |b_v - b_v'|^q + \|\theta^{\to v} - (\theta')^{\to v}\|_q^q \right)$$
$$+ \boldsymbol{\|\Phi(\theta)\|_q^q} \max_{p \in \mathcal{P} : p_{\text{end}} \notin N_{\text{in}}} \sum_{\ell=1}^{\texttt{length}(p)-1} \left( |b_{p_\ell} - b_{p_\ell}'|^q + \|\theta^{\to p_\ell} - (\theta')^{\to p_\ell}\|_q^q \right).$$

The term in **blue** can be replaced by $\mathbf{\min\left(\|\Phi(\theta)\|_q^q, \|\Phi(\theta')\|_q^q\right)}$ by repeating the proof with $\theta$ and $\theta'$ exchanged (everything else is invariant under this exchange). $\qquad\square$

**Lemma F.3.** *Consider a DAG ReLU network with $L := D - 1$ where the depth $D$ is $\max_{\text{path } p \in \mathcal{P}} |\texttt{length}(p)|$ and width $W = \max(d_{out}, \max_{\text{neuron } v \in N} |\operatorname{ant}(v)|)$ where $\operatorname{ant}(v)$ is the set of antecedents of $v$ in the DAG. Denote by $\theta$ the normalized parameters of $\theta$ as obtained as the output of Algorithm 1 in (Gonon et al., 2024a) with $q = 1$, i.e., $\theta$ is obtained from $\theta$ by rescaling neurons from the input to output layer, ensuring every neuron has a vector of incoming weights equal to one on all layers except the last one. It holds for every $\theta, \theta'$ and every $q \in [1, \infty)$*

$$\|\Phi(\theta) - \Phi(\theta')\|_q^q \leqslant (W^2 + \min(\|\Phi(\theta)\|_q^q, \|\Phi(\theta')\|_q^q) \cdot LW)\|\theta - \theta'\|_\infty^q$$

Lemma 4.3 corresponds to Lemma F.3 with $q = 1$.

*Proof of Lemma F.3.* Lemma B.1 of (Gonon et al., 2024a) guarantees that $\Phi(\texttt{N}(\theta)) = \Phi(\theta)$ for every $\theta$. In particular,

$$\|\Phi(\theta) - \Phi(\theta')\|_1 = \|\Phi(\texttt{N}(\theta)) - \Phi(\texttt{N}(\theta'))\|_1$$

so it is enough to prove Lemma F.3 for *normalized* parameters, so we may and will assume $\theta = \texttt{N}(\theta), \theta' = \texttt{N}(\theta')$. Denote

$\bar{\theta}^{\to v} := (\theta^{\to v}, b_v)$. With this notation, (18) implies (for normalized parameters $\theta, \theta'$)

$$\|\Phi(\theta) - \Phi(\theta')\|_q^q \leqslant \sum_{v \in N_{\text{out}} \backslash N_{\text{in}}} \|\bar{\theta}^{\to v} - (\bar{\theta}')^{\to v}\|_q^q + \min(\|\Phi(\theta)\|_q^q, \|\Phi(\theta')\|_q^q) \cdot \max_{p \in \mathcal{P}: p_{\text{end}} \notin N_{\text{in}}} \sum_{\ell=1}^{\text{length}(p)-1} \|\bar{\theta}^{\to p_\ell} - (\bar{\theta}')^{\to p_\ell}\|_q^q$$

$$\leqslant \sum_{v \in N_{\text{out}} \backslash N_{\text{in}}} |\operatorname{ant}(v)| \cdot \|\bar{\theta}^{\to v} - (\bar{\theta}')^{\to v}\|_\infty^q$$

$$+ \min(\|\Phi(\theta)\|_q^q, \|\Phi(\theta')\|_q^q) \cdot \max_{p \in \mathcal{P}: p_{\text{end}} \notin N_{\text{in}}} \sum_{\ell=1}^{\text{length}(p)-1} |\operatorname{ant}(p_\ell)| \cdot \|\bar{\theta}^{\to p_\ell} - (\bar{\theta}')^{\to p_\ell}\|_\infty^q$$

$$\leqslant \left( \sum_{v \in N_{\text{out}} \backslash N_{\text{in}}} |\operatorname{ant}(v)| + \min(\|\Phi(\theta)\|_q^q, \|\Phi(\theta')\|_q^q) \cdot \max_{p \in \mathcal{P}: p_{\text{end}} \notin N_{\text{in}}} \left( \sum_{\ell=1}^{\text{length}(p)-1} |\operatorname{ant}(p_\ell)| \right) \right) \|\theta - \theta'\|_\infty^q$$

The maximum length of a path is $D = L + 1$. Moreover $W \geqslant d_{\text{out}} = |N_{\text{out}}|$ and $W \geqslant |\operatorname{ant}(v)|$ for every neuron, so this yields

$$\|\Phi(\theta) - \Phi(\theta')\|_q^q \leqslant (W^2 + \min(\|\Phi(\theta)\|_q^q, \|\Phi(\theta')\|_q^q) \cdot LW)\|\theta - \theta'\|_\infty.$$

$\square$

## G. Recovering a known bound with Theorem 4.1

It is already known in the literature that for every input $x$ and every parameters $\theta, \theta'$ (even with different signs) of a layered fully-connected neural network with $L$ affine layers and $L + 1$ layers of neurons, $N_0 = N_{\text{in}}, \ldots, N_L = N_{\text{out}}$, width $W := \max_{0 \leqslant \ell \leqslant L} |N_\ell|$, and each matrix having some operator norm bounded by $R \geqslant 1$, it holds (Gonon et al., 2023, Theorem III.1 with $p = q = \infty$ and $D = \|x\|_\infty$)(Neyshabur et al., 2018; Berner et al., 2020):

$$\|R_\theta(x) - R_{\theta'}(x)\|_1 \leqslant (W\|x\|_\infty + 1)WL^2 R^{L-1}\|\theta - \theta'\|_\infty.$$

Can it be retrieved from Theorem 4.1? Next corollary almost recovers it: with $W \max(\|x\|_\infty, 1)$ instead of $W\|x\|_\infty + 1$, and $2L$ instead of $L^2$. This is better as soon as there are at least $L \geqslant 2$ layers and as soon as the input satisfies $\|x\|_\infty \geqslant 1$.

**Corollary G.1.** *(Gonon et al., 2023, Theorem III.1) Consider a simple layered fully-connected neural network architecture with $L \geqslant 1$ layers, corresponding to functions $R_\theta(x) = M_L \operatorname{ReLU}(M_{L-1} \ldots \operatorname{ReLU}(M_1 x))$ with each $M_\ell$ denoting a matrix, and parameters $\theta = (M_1, \ldots, M_L)$. For a matrix $M$, denote by $\|M\|_{1,\infty}$ the maximum $\ell^1$-norm of a row of $M$. Consider $R \geqslant 1$ and define the set $\Theta$ of parameters $\theta = (M_1, \ldots, M_L)$ such that $\|M_\ell\|_{1,\infty} \leqslant R$ for every $\ell \in [\![1, L]\!]$. Then, for every parameters $\theta, \theta' \in \Theta$, and every input $x$:*

$$\|R_\theta(x) - R_{\theta'}(x)\|_1 \leqslant \max(\|x\|_\infty, 1)2LW^2 R^{L-1}\|\theta - \theta'\|_\infty.$$

*Proof.* For every neuron $v$, define $f(v) := \ell$ such that neuron $v$ belongs to the output neurons of matrix $M_\ell$ (i.e., of layer $\ell$). By Lemma F.1 with $q = 1$, we have for every neuron $v$

$$\|\Phi^{\to v}(\theta) - \Phi^{\to v}(\theta')\|_1$$

$$\leqslant \max_{p \in \mathcal{P}^{\to v}} \sum_{\ell=1}^{\text{length}(p)} \left( \prod_{\substack{k=\ell+1 \\ \leqslant \|M_{f(p_k)}\|_{1,\infty} \\ \leqslant R}}^{\text{length}(p)} \underbrace{\|\theta^{\to p_k}\|_1}_{} \right)$$

$$\left( \underbrace{|b_{p_\ell} - b'_{p_\ell}|}_{=0 \text{ (no biases)}} + \underbrace{\|\theta^{\to p_\ell} - (\theta')^{\to p_\ell}\|_1}_{\leqslant |\operatorname{ant}(p_\ell)|\|\theta-\theta'\|_\infty \leqslant W\|\theta-\theta'\|_\infty} \max_{u \in \operatorname{ant}(p_\ell)} \|\Phi^{\to u}(\theta')\|_1 \right) \tag{19}$$

$$\leqslant W\|\theta - \theta'\|_\infty \max_{p \in \mathcal{P}^{\to v}} \sum_{\ell=1}^{\text{length}(p)} R^{\text{length}(p)-\ell} \max_{u \in \operatorname{ant}(p_\ell)} \|\Phi^{\to u}(\theta')\|_1 \tag{20}$$

with the convention that an empty sum and product are respectively equal to zero and one. Consider $\theta' = 0$. It holds $\|\Phi^{\to u}(\theta')\|_1 = 0$ for every $u \notin N_{\text{in}}$, and $\|\Phi^{\to u}(\theta')\|_1 = 1$ for input neurons $u$ (Definition A.5). Therefore, we have:

$$\max_{u \in \text{ant}(p_\ell)} \|\Phi^{\to u}(\theta')\|_1 = \mathbb{1}_{\text{ant}(p_\ell) \cap N_{\text{in}} \neq \emptyset} = \mathbb{1}_{\ell=1 \text{ and } p_0 \in N_{\text{in}}}. \tag{21}$$

Specializing Inequality (19) to $\theta' = 0$ and using Equation (21) yields

$$\|\Phi^{\to v}(\theta)\|_1 \leqslant \max_{p \in \mathcal{P}^{\to v}} \sum_{\ell=1}^{\text{length}(p)} \underbrace{\left( \prod_{k=\ell+1}^{\text{length}(p)} R \right)}_{} \underbrace{\|\theta^{\to p_\ell}\|_1}_{\substack{\leqslant \|M_{f(p_\ell)}\|_{1,\infty} \\ \leqslant R}} \underbrace{\max_{u \in \text{ant}(p_\ell)} \|\Phi^{\to u}(\theta')\|_1}_{= \mathbb{1}_{\ell=1 \text{ and } p_0 \in N_{\text{in}}}}$$

$$= \max_{p \in \mathcal{P}^{\to v}: p_0 \in N_{\text{in}}} R^{\text{length}(p)}. \tag{22}$$

Since the network is layered, every neuron $u \in \text{ant}(p_\ell)$ is on the $\ell-1$-th layer, and every $p' \in \mathcal{P}^{\to u}$ is of length $\ell-1$, hence we deduce using Inequality (20), Equation (22) for $\theta'$ and $u$:

$$\|\Phi^{\to v}(\theta) - \Phi^{\to v}(\theta')\|_1 \leqslant W\|\theta - \theta'\|_\infty \max_{p \in \mathcal{P}^{\to v}} \sum_{\ell=1}^{\text{length}(p)} R^{\text{length}(p)-\ell} \underbrace{\max_{u \in \text{ant}(p_\ell)} \max_{p' \in \mathcal{P}^{\to u}: p_0' \in N_{\text{in}}} R^{\text{length}(p')}}_{= R^{\ell-1}}$$

$$= W\|\theta - \theta'\|_\infty \max_{p \in \mathcal{P}^{\to v}} \underbrace{\sum_{\ell=1}^{\text{length}(p)} R^{\text{length}(p)-1}}_{\leqslant L R^{L-1}}$$

$$\leqslant L W R^{L-1} \|\theta - \theta'\|_\infty.$$

We get:

$$\|\Phi(\theta) - \Phi(\theta')\|_1 = \sum_{v \in N_{\text{out}} \setminus N_{\text{in}}} \|\Phi^{\to v}(\theta) - \Phi^{\to v}(\theta')\|_1$$

$$\leqslant |N_{\text{out}} \setminus N_{\text{in}}| \cdot L W R^{L-1} \|\theta - \theta'\|_\infty$$

$$\leqslant L W^2 R^{L-1} \|\theta - \theta'\|_\infty. \tag{23}$$

Using Corollary B.2 with $q = 1$, we deduce that as soon as $\theta, \theta'$ satisfy $\theta_i \theta_i' \geqslant 0$ for every parameter coordinate $i$, then for every input $x$:

$$\|R_\theta(x) - R_{\theta'}(x)\|_1 \leqslant \max(\|x\|_\infty, 1) L W^2 R^{L-1} \|\theta - \theta'\|_\infty. \tag{24}$$

Now, consider general parameters $\theta$ and $\theta'$. Define $\theta^{\text{inter}}$ to be such that for every parameter coordinate $i$:

$$\theta_i^{\text{inter}} = \begin{cases} \theta_i' & \text{if } \theta_i \theta_i' \geqslant 0, \\ 0 & \text{otherwise.} \end{cases}$$

By definition, it holds for every parameter coordinate $i$: $\theta_i^{\text{inter}} \theta_i \geqslant 0$ and $\theta_i^{\text{inter}} \theta_i' \geqslant 0$ so we can apply Inequality (24) to the pairs $(\theta, \theta^{\text{inter}})$ and $(\theta^{\text{inter}}, \theta')$ to get:

$$\|R_\theta(x) - R_{\theta'}(x)\|_1 \leqslant \|R_\theta(x) - R_{\theta^{\text{inter}}}(x)\|_1 + \|R_{\theta^{\text{inter}}}(x) - R_{\theta'}(x)\|_1$$

$$\leqslant \max(\|x\|_\infty, 1) L W^2 R^{L-1} \left( \|\theta - \theta^{\text{inter}}\|_\infty + \|\theta^{\text{inter}} - \theta'\|_\infty \right).$$

It remains to see that $\|\theta - \theta^{\text{inter}}\|_\infty + \|\theta^{\text{inter}} - \theta'\|_\infty = 2\|\theta - \theta'\|_\infty$. Consider a parameter coordinate $i$.

If $\theta_i \theta_i' \geqslant 0$ then $\theta_i^{\text{inter}} = \theta_i'$ and:

$$|\theta_i - \theta_i'| = |\theta_i - \theta_i^{\text{inter}}| + |\theta_i^{\text{inter}} - \theta_i'|.$$

Otherwise, $\theta_i^{\text{inter}} = 0$ and:

$$|\theta_i - \theta_i'| = |\theta_i| + |\theta_i'|$$
$$= |\theta_i - \theta_i^{\text{inter}}| + |\theta_i^{\text{inter}} - \theta_i'|.$$

This implies $\|\theta - \theta^{\text{inter}}\|_\infty = \max_i |\theta_i - \theta_i^{\text{inter}}| \leqslant \max_i |\theta_i - \theta_i^{\text{inter}}| + |\theta_i^{\text{inter}} - \theta_i'| = \|\theta - \theta'\|_\infty$ and similarly $\|\theta^{\text{inter}} - \theta'\|_\infty \leqslant \|\theta - \theta'\|_\infty$. This yields the desired result:

$$\|R_\theta(x) - R_{\theta'}(x)\|_1 \leqslant \max(\|x\|_\infty, 1) 2LW^2 R^{L-1} \|\theta - \theta'\|_\infty. \qquad \square$$

