# OpenReview forum: "A Rescaling-Invariant Lipschitz Bound Based on Path-Metrics for Modern ReLU Network Parameterizations"
_ICML.cc/2025/Conference — ICML 2025 poster_

### Official Review · Reviewer_XzWf · 2025-03-08

**Overall Recommendation:** 1

**Summary:**

The paper proves a new reparameterization invariant Lipschitz bound in terms of the “path-metrics” of the parameters. The bound applies generally to network architectures with pooling and skip connections. Using the bound, the authors propose a rescaling-invariant pruning criterion.

**Claims And Evidence:**

The authors claim that their bound is rescaling invariant, while it is theoretically sound due to the construction of path-lifting space, there is no experimental evidence that the bound is non-vacuous, e.g. both sides of the bound correlate with one another.

Line 311-315 left column: “We show this to match the accuracy of magnitude pruning when applied to ResNets trained on Imagenet in the lottery ticket context (Frankle et al., 2020), while being rescaling-invariant”. I cannot find any figures/tables that support this claim, except what is briefly mentioned in line 396-399.

**Essential References Not Discussed:**

No work absolutely needs to be brought up. But I think work related to rescaling invariant sharpness is worth mentioning due to the reason above. Similarly, more work that shows the usefulness of a Lipschitz bound can be mentioned.

**Experimental Designs Or Analyses:**

There is little or no experimental result shown.

**Methods And Evaluation Criteria:**

There is no empirical evaluation.

**Other Comments Or Suggestions:**

No other comments.

**Other Strengths And Weaknesses:**

For the main inequality (3), if we have a normalization layer on the input (which is valid according to line 77), wouldn’t it make the inequality vacuous, since we can arbitrarily scale the input with R and \theta stays the same?

**Questions For Authors:**

No other questions.

**Relation To Broader Scientific Literature:**

The authors claim that they are the first in literature that propose a scale invariant Lipschitz bound. I found the support for the usefulness of a Lipschitz bound weak in the paper, and only two papers are mentioned (Neyshabur et al., 2018; Gonon et al, 2023). Moreover, the Lipschitz bound in Neyshabur et al., 2018 is only used to bound sharpness, which in turn connects to generalization. On the sharpness side, there are plenty of rescaling invariant bound proposed since Dinh et al., 2017, and I feel that the authors should more thoroughly discuss this connection with sharpness bounds, and see how their bound is relevant. In fact, if MSE loss is used, sharpness is exactly the norm of gradient of the output of networks w.r.t. the parameters (see e.g. Wen et al., 2023, lemma 4.1; Ma & Ying, 2021, Equation 3).

**Theoretical Claims:**

I have checked the proof sketch of theorem 3.1, and it looks sound to me.

---

> ### Author Rebuttal · Authors · 2025-03-31
>
> Thank you for your review. We address your points below.
>
> 1. > what if there is a normalization layer on the input
>
> We assume you refer to batch normalization. As detailed in Gonon et al. 2024, batch normalization layers *as they behave at inference* are indeed covered in the path-lifting framework. While at training batch normalization dynamically adapts its weights to batches, this is no longer the case at inference where it acts as a plain affine layer. As a result, *the problem you raise does not appear*. The revision will make it clear.
>
>
> 2. > mentioning the relation to sharpness
>
> Sharpness measures [1, 2] are typically defined in terms of (averaged) loss differences  $L(\theta+\delta) - L(\theta)$ over perturbations $\delta\in B(0,r)$ around a local minima $\theta$. As such, they are not rescaling-invariant measures of $\theta$ *alone*, since rescaling only $\theta$ but not the perturbation $\delta$ will change this quantity in general.  *One* consequence of our main Theorem 3.1 is to further bound this type of sharpness measure  (again by something which is not invariant in $\theta$ alone, but invariant in both $\delta,\theta$). Indeed, if $L(\theta) = \sum_{i=1}^n \ell(R_{\theta}(x_i), y_i)$ with $\ell$ Lipschitz in its first argument (e.g., cross-entropy, or squared loss on a compact), it holds $\|\ell(R_{\theta+\delta}(x_i), y_i) - \ell(R_{\theta}(x_i), y_i)\| \leq c \|\|R_{\theta+\delta}(x_i) - R_{\theta}(x_i)\|\|$ and the latter is bounded by $c \|x_i\| \|\|\Phi(\theta+\delta) - \Phi(\theta)\|\|$ according to Theorem 3.1.
> We will add this to the final version.
>
> 3. > I found the support for the usefulness of a Lipschitz bound weak in the paper
>
> If we understood correctly, you are asking about the usefulness of Lipschitz bounds beyond their use to bound sharpness as above. The revision will cite additional papers [3-9] that use *non-invariant* Lipschitz bounds of the same type as those in (Neyshabur et al., 2018; Gonon et al, 2023) to design new algorithms and guarantees on pruning, quantization and generalization (not through sharpness, but via covering numbers). In all of these papers, Lipschitzness comes in as a crucial property to control how the function changes with small weight perturbations. However, these papers unfortunately suffer from the two problems motivating our paper: 1) they use non-invariant bounds, which not only can be made arbitrarily pessimistic but that might also yield algorithms with huge performance drops when run on rescaling-equivalent parameters (Figure 6), and 2) they only hold for simple fully-connected models organized in layers.
>
>
> [1] How Does Sharpness-Aware Minimization Minimize Sharpness? Wen et al. 2023. Table 1.
>
> [2] A Modern Look at the Relationship between Sharpness and Generalization. Andriushchenko et al. 2023. Equation (1).
>
> [3] Liebenwein et al., Provable Filter Pruning for Efficient Neural Networks, ICLR 2020.
>
> [4] Baykal et al., Data-dependent Coresets for Compressing Neural Networks with Applications to Generalization Bounds, ICLR 2019.
>
> [5] SiPPing Neural Networks: Sensitivity-informed Provable Pruning of Neural Networks. Baykal et al. 2019.
>
> [6] Arora et al., Stronger Generalization Bounds for Deep Nets via a Compression Approach, ICML 2018.
>
> [7] Zhang et al., Post-training Quantization for Neural Networks with Provable Guarantees, 2023.
>
> [8] Lybrand and Saab, A Greedy Algorithm for Quantizing Neural Networks, JMLR 2021.
>
> [9] Schnoor et al., Generalization Error Bounds for Iterative Recovery Algorithms Unfolded as Neural Networks, 2022.

---

> > ### Comment · Reviewer_XzWf · 2025-04-02
> >
> > I meant layer normalization. I see that you wrote batch normalization instead, so maybe it does not apply to your setting.
> >
> > [2] above uses rescaling invariant sharpness measure and is invariant under multiplicative reparametrizations. The explanation immediately follows equation (1) therein. I don't understand why the authors say otherwise.
> >
> > There are many scale invariant definitions of sharpness:
> >
> > 1. Tsuzuku, Y., Sato, I., and Sugiyama, M. Normalized flat minima: Exploring scale invariant definition of flat minima for neural networks using pac-bayesian analysis,2019.
> >
> > 2.Kwon, J., Kim, J., Park, H., and Choi, I. K. Asam: Adaptive
> > sharpness-aware minimization for scale-invariant learning of deep neural networks. In International Conference
> > on Machine Learning, pp. 5905–5914. PMLR, 2021. (Used in [2])
> >
> > 3.Rangamani, Akshay, et al. "A scale invariant flatness measure for deep network minima." arXiv preprint arXiv:1902.02434 (2019).
> >
> > I lean towards rejection also because there is no proper exhibition of experimental results in the main text, and no comparison to other pruning methods. All I see in the main text is Table 3, which shows the computing time for the pruning methods. The authors should put Table 4 and Figure 6 in the main text.
> >
> > Also, the pruning method is not a direct evaluation of how tight the bound in theorem 3.1 is. The authors should evaluate both sides of the inequality to see if the inequality is useful.

---

> > > ### Author Response · Authors · 2025-04-02
> > >
> > > **Layer normalization**: indeed, normalization layers are not rescaling-invariant, and as such are not covered by the path-lifting framework, unlike batch normalization layers.
> > >
> > > **Experimental results in the main text**: in the final version we can group existing figures to gain space and move relevant aspects of appendix D to the main text to complete Table 3.
> > >
> > > **Rescaling-invariant sharpness measures**: indeed we based our answer on the definition of sharpness from [1] and missed the adaptiveness allowed by the additional vector $c$ from [2] and the other references you point out, which yield rescaling-invariant measures, thank you.
> > >
> > > **Pros and cons of the proposed Lipschitz bound**: Table 2 summarizes key properties of various pruning criteria including the one directly derived from our Lipschitz bound. A similar table can be carved out for other potential applications of the bound such as sharpness measures. We will include it to highlight the pros and cons of the bound: while the pruning criteria and sharpness measures derived from it may be legitimately criticized for their potential numerical sub-optimality for such or such application, one of the main strengths of the bound (beyond its generic rescaling invariance) is also its flexible applicability to diverse settings, thanks to its independence from a particular dataset or a particular loss.

---

### Official Review · Reviewer_duh5 · 2025-03-10

**Overall Recommendation:** 3

**Summary:**

The paper derives a Lipschitz upper bound for neural networks with ReLU and k-max-pooling activations. For two parameters $\Theta$ and $\Theta'$, the paper shows that $||R_{\Theta}(x)-R_{\Theta'}(x)||_1\leq max(||x||_∞,1) ||\Phi(\Theta)-\Phi(\Theta')||_1$, with an assumption that $\mathrm{sign}(\Theta)=\mathrm{sign}(\Theta')$.


Here the new vector $\Phi(\Theta)$ is the lifting of the original parameter $\Theta$. This upper bound is rescaling-invariant, meaning that the upper bound only relies on the intrinsic property of the network, instead of the parameter. For any two parameters $\Theta_1$ and $\Theta_2$ such that $R_{\Theta_1}=R_{\Theta_2}$, then their liftings $\Phi(\Theta_1)=\Phi(\Theta_2)$.

This inequality can be used to prune a dense network into a sparse one, while ensuring it has similar performance compared with the dense network.

**Claims And Evidence:**

Yes

**Essential References Not Discussed:**

I didn't discover this issue.

**Experimental Designs Or Analyses:**

Didn't find any issue.

**Methods And Evaluation Criteria:**

Yes

**Other Comments Or Suggestions:**

see 'Questions For Authors'.

**Other Strengths And Weaknesses:**

Strength: The paper introduces a scaling invariant Lipschitz upper bound using parameter lifting. It is very clear and rigorous. The proof is well written.

Weakness: The main result theorem 3.1 in this paper seems quite simple. The generalization bound derived from this theorem is not clearly stated and proved. (I am actually curious about that.) This theorem 3.1 has an application in pruning, but I am not very certain whether this theorem can be very impactful in other applications.

**Questions For Authors:**

1. The assumption that $\mathrm{sign}(\Theta)=\mathrm{sign}(\Theta')$ seems very strong. If we no longer assume this, is it possible to derive a looser upper bound? If we assume there are only a few edges $i$ such that the weights $\Theta_i\Theta'_i<0$, what bound can you get?

2. Without requiring rescaling invariance, is it possible to extend the activation to leaky ReLU or some smooth activation functions? In that case, what is your upper bound?

3. Is it possible that when the network is very deep and over-parameterized, the output of the network $R_{\Theta}(x)$ is small, but every entry in your lifting $\Phi(\Theta)$ is very large ($\Phi_p(\Theta)\gg0$), so you do not get a meaningful bound when you do pruning using equation (13)? In this case, is it possible to do some pruning while ensuring the output $R_{\Theta}(x)$ doesn't change much?

I will adjust the rating based on the answers and also the comments from other reviewers.

**Relation To Broader Scientific Literature:**

It is a deep learning theory paper that might be useful for pruning.

**Theoretical Claims:**

Yes, the proof of the main theorem 3.1 is correct

---

> ### Author Rebuttal · Authors · 2025-03-31
>
> Thank you for your review. We address your points below.
>
> 1. > assumption of sign consistency and extension to cases where only a few edges have different signs?
>
> As shown by the example in Figure 5, page 13 (that we will move to the main text), the sign assumption cannot be simply removed in Theorem 3.1. This is thus not a limitation of the approach but a limitation of the achievable Lipschitz bound. We will highlight this fact, which is indeed a contribution. It is not difficult to design variants of the counterexample of Figure 5 with network parameters sharing the same sign on every edge except two, by "prepending" and "appending" arbitrary networks to the example.
> Besides, we highlight that this is not a limitation in practice for applications even beyond pruning (for example quantization), and it allows one to obtain generalization bounds with *free* signs (the proof sketch is given line 202, and could certainly be detailed in the supplementary in the final version if requested).
>
> 2. extension to networks with activations beyond the ReLU
>
> The leaky ReLU remains rescaling invariant and piecewise linear. Although the current path-lifting framework used to derive Theorem 3.1 does not cover the leaky ReLU, it is possible that a complete re-examination of this framework might allow for a generalisation of Theorem 3.1 to this activation. Regarding non rescaling-invariant activations, we cannot expect any bound of the same type as in Theorem 3.1 to directly hold as the right-hand-side is rescaling invariant.
>
> 3. meaningful bound when deep/overparameterized networks, as path-coefficients are large
>
> You are perfectly right: although the provided bound is the sharpest of its kind, it somehow remains a worst case over all weights with a prescribed path-norm. It now raises the challenge of obtaining tighter "average" bounds.

---

> > ### Comment · Reviewer_duh5 · 2025-04-07
> >
> > I will adjust my rating after studying the comments from other reviewers carefully.

---

### Official Review · Reviewer_NY8v · 2025-03-12

**Overall Recommendation:** 4

**Summary:**

This paper introduces a novel Lipschitz bound for modern ReLU neural networks that is invariant under neuron-wise rescaling transformations. The key idea is to leverage a "path-lifting" function which transforms the network parameters into a high-dimensional path space, where each coordinate corresponds to the product of weights along a path. While the path‑lifting function and the associated path‑norm have been introduced in previous works [1], this paper extends these ideas by establishing a rescaling‐invariant Lipschitz bound and deriving a practical pruning criterion. In other words, it is not the tool itself that is new, but its effective integration into a Lipschitz analysis framework that remains invariant under neuron‐wise rescaling and its subsequent application.
And the paper further illustrates the utility of this invariant bound by deriving a new rescaling-invariant pruning criterion termed “Path-Magnitude Pruning,” and reports experiments demonstrating that this approach maintains performance under adversarial rescaling that would affect conventional magnitude pruning.

[1] Gonon, A., Brisebarre, N., Riccietti, E., & Gribonval, R. (2023). A path-norm toolkit for modern networks: consequences, promises and challenges. arXiv preprint arXiv:2310.01225.

## Update after rebuttal
 - I thank the authors for their response, which generally resolved my concerns. I will maintain my Overall Recommendation for the manuscript, standing on the acceptance side.

**Claims And Evidence:**

1. Novel Lipschitz Bound and Rescaling-Invariance:
The paper introduces a new Lipschitz upper bound based on path-metrics that is invariant to neuron-wise rescaling. The authors provide mathematical proofs (e.g., Theorem 3.1 and supporting lemmas) demonstrating that, under the assumption that parameter pairs maintain the same sign, the derived bound is both tighter and robust compared to traditional bounds based on standard ℓₚ norms.

2. Path-Magnitude Pruning Criterion:
The paper proposes a pruning criterion (Path-Mag) derived from the proposed Lipschitz bound. The authors offer both an analytical justification (Lemma 4.2) and experimental results on ResNet-18 trained on ImageNet, demonstrating that the approach maintains performance under random rescaling—a scenario where conventional magnitude pruning fails.

3. The major limitation is the assumption of parameters. Although practically justifiable in many scenarios (e.g., during pruning or with small gradient steps), this condition restricts the generality of the claim. Additionally, empirical evidence is limited to a single network (ResNet-18), leaving some uncertainty about applicability to other architectures.

**Essential References Not Discussed:**

I did not identify any major omissions of essential references. The manuscript appears to adequately cover the relevant literature.

**Experimental Designs Or Analyses:**

The experimental design and analyses are valid and sufficiently support the theoretical claims. The approach is well-motivated and the comparative evaluation is clear, despite being limited to a single network architecture (ResNet-18) and lacking extensive hyperparameter exploration.

**Methods And Evaluation Criteria:**

The proposed methods and evaluation criteria are appropriate for obtaining a rescaling-invariant Lipschitz bound based on path-lifting functions.

**Other Comments Or Suggestions:**

- **Further Discussion on Assumptions:**
  The authors should include more discussion on the practical impact of the sign consistency condition and potential extensions when this assumption is relaxed.


- **Enhanced Experimental Comparisons:**
  It would be beneficial to include additional comparisons with other norm-based or invariance-aware methods, especially under diverse conditions of weight perturbation.

**Other Strengths And Weaknesses:**

### Strengths

- **Innovative Theoretical Approach:**
  The use of path-lifting to achieve a rescaling-invariant Lipschitz bound is a clever idea that addresses a long-standing limitation in norm-based bounds.

- **Broad Applicability:**
  The theoretical results extend to modern network architectures beyond simple feedforward models, encompassing pooling layers and skip connections.

- **Practical Utility:**
  The derived bound is directly applied to design a rescaling-invariant pruning method, and experimental results confirm its practical benefits.

### Weaknesses

- **Assumption Limitations:**
  The theoretical guarantees require that corresponding parameters share the same sign, which might limit applicability in some practical settings.


- **Clarity in Presentation:**
There are some typos, for example, in Definition A.2, the definition of $\theta^{v\rightarrow}$ appears to be incorrect.

**Questions For Authors:**

No other comments.

**Relation To Broader Scientific Literature:**

This work contributes to the broader literature in several ways:

- **Theoretical Insight:** It builds on and extends prior work on the path-norm and Lipschitz analysis of neural networks (e.g., by Neyshabur et al., Gonon et al.).

- **Practical Applications:** The method provides a new tool for pruning and potentially quantization, addressing known issues with conventional parameter norm bounds.

- **General Applicability:** It successfully generalizes to modern network architectures that include pooling and skip connections, areas where traditional bounds are less effective.

Overall, the contribution is well situated within the existing literature on neural network generalization and robustness.

**Theoretical Claims:**

Overall, the proofs for the primary theoretical claims—including Theorem 3.1 (and its extensions) and the supporting lemmas—are correct and employ rigorous mathematical reasoning. The primary concern lies in the necessary assumption of sign consistency, which, although justifiable in many practical applications, narrows the scope of the theoretical claims. Furthermore, some steps in the proofs could benefit from enhanced clarity to assist readers less familiar with the underlying techniques.

---

> ### Author Rebuttal · Authors · 2025-03-31
>
> Thank you for your review. We address your points below.
>
> 1. > The major limitation is the assumption of parameters/assumption of sign consistency
>
> As shown by the example in Figure 5, page 13 (that we will move to the main text), the sign assumption cannot be simply removed in Theorem 3.1. This is thus not a limitation of the approach but a limitation of the achievable Lipschitz bound. We will highlight this fact, which is indeed a contribution.
> Besides, we highlight that this is not a limitation in practice for applications to quantization and pruning, and that it allows to obtain generalization bounds with *free* signs.
>
> 2. > limited empirical evidence
>
> The pruning experiment on a ResNet-18 is intended to be a proof-of-concept illustration of *one* possible application of our main contribution, which is theoretical: the nontrivial proof of Theorem 3.1. To avoid unnecessary energy consumption, we voluntarily avoided extensive comparisons which are clearly out of the scope of our claimed research contribution. In fact, the code that we will release on a non-anonymous repository allows to apply the same approach to more than 37 architectures available on torch.

---

### Decision · Program_Chairs · 2025-05-01

**Decision:**

Accept (poster)

**Comment:**

This article presents bounds on the Lipschitz constant of a ReLU network that is invariant to multiplicative reparameterizations. The core construction is based on a lifting to "path space," in which the output of the network is written as a sum over paths from input to output of a product of weights along the path. Some empirical applications to pruning and robustness are presented. While not extensive, they are reassuring.